# Epigenetic gene regulation is controlled by distinct regulatory complexes utilizing specialized paralogs of TELOMERE REPEAT BINDING FACTORS

**Maik Mendler** [ID][⊕], **Kristin Krause**[⊕][¤], **Simone Zündorf, Prathamesh Sannak**[ID]**,**
**Petra Tänzler, Sara Stolze**[ID]**, Hirofumi Nakagami**[ID]**, Franziska Turck**[ID]*

Department of Plant Developmental Biology, Max Planck Institute for Plant Breeding Research, Köln, Germany

⊕ These authors contributed equally to this work.
¤ Current address: Illumina Solutions Center, Berlin, Germany
* turck@mpipz.mpg.de

## Abstract

Epigenetic regulators shape chromatin landscapes, allowing cells to express distinct gene sets depending on cell-type, developmental stage or environmental cues. These regulatory complexes rely on interactions with sequence-specific DNA binding proteins, such as the small family of TELOMERE REPEAT BINDING FACTORS (TRBs). TRBs are components of chromatin regulatory complexes with opposing functions, such as the epigenetic repressors Polycomb Repressive Complex 2 (PRC2) and a JMJ14/NAC complex that respectively add and removes the repressive H3K27me3 and positive H3K4me3 modification, but also with the plant-specific PEAT complex that is linked to histone acetylation and gene activation. We dissected the partial redundancy between TRB1, TRB2 and TRB3 in target gene selection and interaction with different chromatin regulatory complexes. High redundancy of TRBs is suggested by major phenotypic changes that are only observed *trb* triple mutants; however, we found different target site preference between TRB1-3 and preferred partnership with chromatin complexes. Furthermore, TRB paralogs interacted with the NuA4 histone acetylation complex, both together with and in absence of PEAT. Among the three paralogs, TRB1 had more unique binding sites and correlated stronger with PEAT and NuA4 functions. In contrast, TRB2 and TRB3 were more dependent on the presence of *bona fide* telo-box motifs and were more likely to be found at PRC2 associated sites. Overall, we provide insight into the diverse roles of TRBs in epigenetic gene regulation and how their diversification contributes to their apparent redundancy, as well as their observed activating and repressing effects on gene expression.

which permits unrestricted use, distribution, and reproduction in any medium, provided the original author and source are credited.

**Data availability statement:** The ChIPseq and RNAseq data that support the findings of this study are publicly available from the European Nucleotide Archive (ENA) with the identifier PRJEB63124. The proteomics data that support the findings of this study are publicly available from the Proteome Xchange data base with the identifier PXD069673. Scripts for read processing, alignments, and downstream data analysis and visualizations are available in Dryad (https://doi.org/10.5061/dryad.gtht76j2r).

**Funding:** This work was supported by the Max Planck Society(MM. KK, SZ, PS, PT, SS, HN, FT) and by the DFG (CEPLAS, EXC 2048/1 Project ID: 390686111 to FT). The funders had no role in study design, data collection and analysis, decision to publish, or preparation of the manuscript.

**Competing interests:** The authors have declared that no competing interests exist.

## Author summary

By controlling access to DNA, epigenetic regulators enable cells to select different gene sets for expression or repression depending on the cell type, developmental stage, or environmental cues. These regulatory complexes rely on interactions with DNA-binding proteins to locate their target regions. Interestingly, a small family of plant DNA-binding proteins, originally identified as telomere-binding factors (TRBs), interacts with both activating and repressing epigenetic regulators, suggesting a role in maintaining the balance between gene activation and repression. We mapped the genome-wide binding of individual family members and compared our results with those of their interacting partners, for which data were available. We found a clear separation between the binding sites for active and repressive complexes, although some genes were associated with several TRB associated complexes. The TRB family is highly redundant, as the presence of even one functional allele can prevent the catastrophic effects on plant development observed in full mutants. Despite this redundancy, we found that family members had very distinct binding site preferences, showing a preference for either repressive or activating epigenetic regulators. Redundancy contributes to the robustness of gene regulation, while specialization allows optimized responses to changing conditions. The extent to which TRBs contribute to both robustness and responsiveness will be a focus of future research.

## Background

In *Arabidopsis thaliana*, telo-box motifs are widely found at gene regulatory elements; furthermore, they are native to the telomeres at chromosomal ends, where they occur as direct repeats (TTTAGGG × n; n = 2–1000+) and associate with several telomere repeat binding proteins, including TELOMERE REPEAT BINDING FACTOR (TRB) 1-5 [1–3]. TRB proteins belong to the Single myb histone (Smh) family and contain an N-terminal myb, a H1/H5-linker and a C-terminal coiled-coil domain [4]. The direct interaction of TRBs with telomere repeats is mediated by their N-terminal myb-domain, which belongs to the telobox class of myb-domains that is shared among telomere repeat binding proteins in all eukaryotes [5,6]. TRBs directly interact with the telomerase subunit TERT and are thought to be part of the plant shelterin complex which aids the telomerase in solving the end-replication problem and protects telomere ends from being falsely recognized as DNA double strand breaks [1,7]. The H1/H5-linker histone domain is involved in the formation of TRB multimers at telomere ends. For TRB1, it was shown that binding to interstitial telomeric repeats, which are relics of chromosomal fusions specific to *A. thaliana*, is prevented by the presence of linker histone H1, indicating that while the H1/H5 domain may contribute to target binding, it cannot outcompete the canonical linker histone [8]. The C-terminal coiled-coil domain is thought to mediate the interaction with other proteins [5,9]. Telo-box motifs were first linked to promoter regions of highly expressed genes

of the translation machinery [3]. Genome-wide binding analysis of TRB1 confirmed the link to genes encoding for the translational apparatus [9,10].

TRB1 is an integral component of the plant-specific PWWP-ENHANCER OF POLYCOMB-LIKE-ARID-TRB (PEAT) complex, which is predominantly involved in gene activation [11,12]. PEAT activates target genes through a dual approach that involves histone acetylation and deubiquitination of mono-ubiquitinated H2A (H2AKub1) [11]. Histone acetylation is facilitated through HISTONE ACETYLASE RELATED TO MYST (HAM) 1 and 2 which are shared components of PEAT and the Nucleosomal Acetyltransferase of histone H4 (NuA4) complex [11]. Co-purification of UBIQUITIN SPECIFIC PROTEASE 5 (UBP5) and PEAT positions PEAT as a direct antagonist of the epigenetic repressor Polycomb Repressive Complex (PRC) 1, which sets the H2AKub1 mark [13].

Mutated alleles of *TRB1* and *TRB3* were identified as genetic enhancers of mutations in *LIKE-HETEROCHROMATIN PROTEIN 1* (*LHP1*) and *CURLY LEAF* (*CLF*). LHP1 and CLF act as accessory and integral part of PRC2, respectively [9,14] and LHP1 also interacts with PRC1 components that catalyze H2AKub1 [15]. PRC2 establishes the covalent modification tri-methylation of histone H3 at lysine 27 (H3K27me3) at thousands of loci, resulting in transcriptional repression of target genes (reviewed by [16,17]). TRBs can recruit PRC2 to telo-box motifs via their direct interactions with the PRC2 components CLF/SWN, which is essential for stable H3K27me3 coverage and epigenetic repression of a subset of these genes [14]. While TRB1 binding sites were under-represented within H3K27me3 marked regions, telo-box motifs were overrepresented, in particular at regions that show reduced H3K27me3 in *clf* mutants [18]. Telo-box motifs were also enriched in regions bound by the PRC2 components FERTILISATION INDEPENDENT ENDSPERM (FIE), SWINGER (SWN) and CLF [19–21]. Finally, TRB1-3 are also part of a transcription repressive complex composed of the histone de-methylase JUMONJI (JMJ14), NAC50 and NAC52 (names after the founding family members No Apical Meristem (NAM1), ARABIDOPSIS TRANSCRIPTION FACTOR (ATAF1/2), CUP SHAPED COTYLEDONE (CUC2)) [22,23].

The association of telo-box motifs and TRBs with repressed as well as highly expressed genes and with repressive as well as activating chromatin complexes raises the question of how the functional context is established and to which extent TRBs are specialized in their molecular function. Phenotypic analysis confirmed the high redundancy between TRB1–3, as strong phenotypic changes were only observed in triple mutants. In contrast, comparison of genome-wide binding revealed many sites exclusively bound by TRB1 that were more likely associated with PEAT/NuA4 or NuA4. In contrast, binding sites preferred by TRB2 and TRB3 over TRB1 were the most highly associated with PRC2-mediated gene repression.

## Results

### Transcriptomic changes in *trb* double and triple mutants indicate a limited partial redundancy between paralogs

As we and others have previously reported, plants homozygous for two mutated alleles of *TRB1*, *TRB2* or *TRB3* are indistinguishable from wild-type controls grown under standard conditions, but a deletion of the third allele has a catastrophic effect on plant development [14,23]. To evaluate the extent of genetic redundancy, we grew plants that still segregated one functional *TRB* allele (*prope triple*) together with all double mutant combinations and Col-0 controls in standard growth conditions (LD, 16h light/8h dark, 21°C). Triple *trb1–2 trb2–3 trb3–2* (*trb123*) mutants were strongly dwarfed and usually died before their *prope triple* or double homozygous siblings flowered (Fig 1A). While *prope triple* mutants that segregated functional alleles of *TRB1* and *TRB3* flowered as controls, plants carrying only one functional allele of *TRB2* flowered significantly earlier indicating that the latter is a slightly weaker paralog with respect to a role in flowering time regulation (Fig 1B).

Both, high redundancy and signs of unequal redundancy, were also observed at the level of transcriptome changes. Using above ground tissue of 14-day-old seedlings grown on soil in standard growth conditions (LD, 16h light/8h dark, 21°C), we found overall 665 differentially expressed genes (DEGs) in *trb* single and double mutants compared to Col-0 (Fig 1C and S1 File). Of these DEGs, most were specific to *trb1* single and *trb1 trb2* double mutants, indicating that *TRB2*

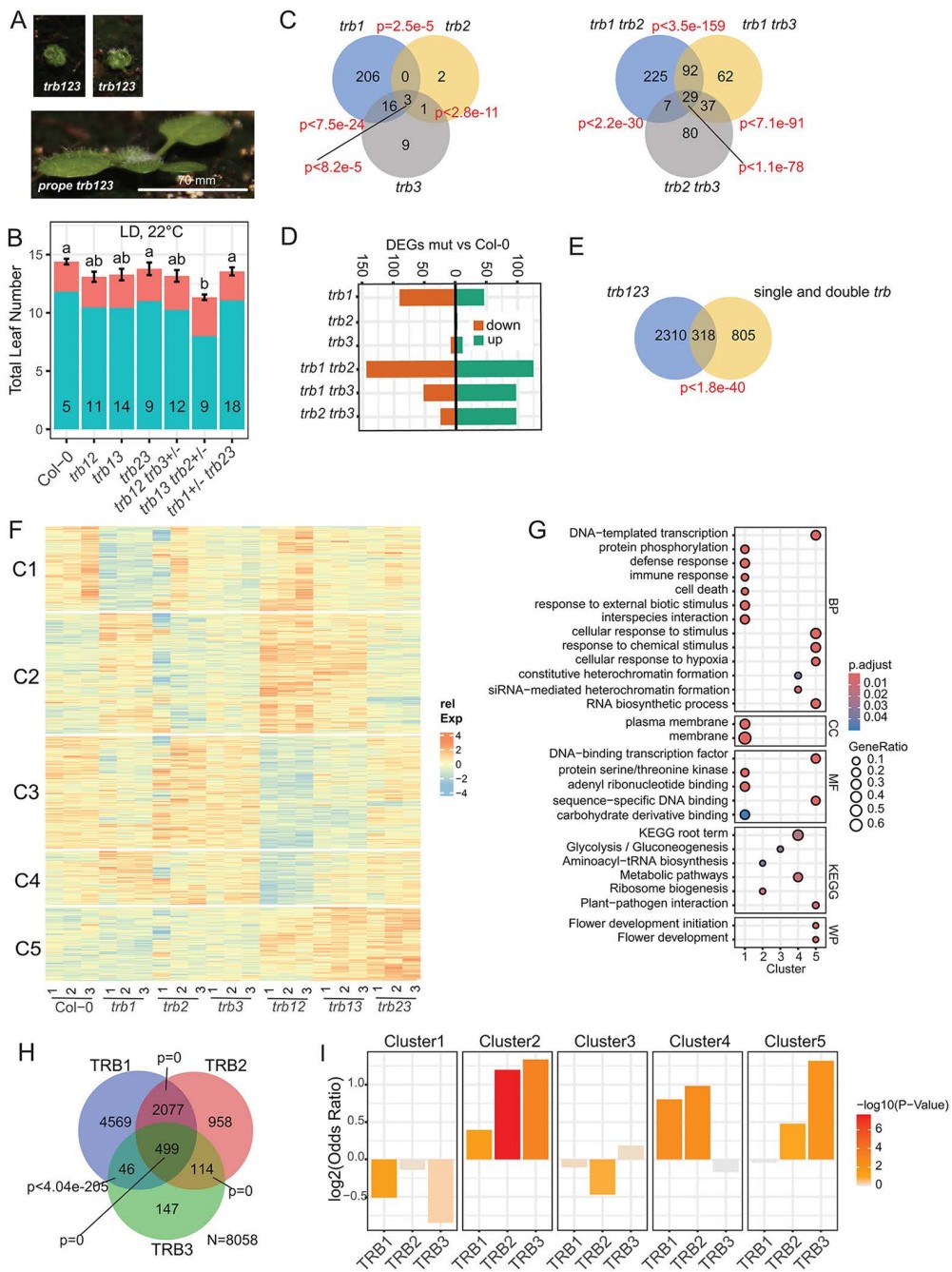

**Fig 1. Limited partial redundancy of TRB1-3. A)** Phenotypes of *trb123* triple mutants and siblings with one functional *TRB* allele. Common scale bar indicated. **B)** Flowering time of double *trb* mutant and *prope triple* mutants scored as total leaves on main shoot (rosette leaves: green, cauline leaves: red). Plants were grown at 22°C in long days (16h light/8h dark) in culture chambers. Significant differences determined by ANOVA, letters indicate HSD groups at p < 0.01. Replicates (5 < n < 18) are indicated within bars. Error bars show standard error of the mean. **C)** Venn diagrams showing the number of differentially expressed genes (DEGs) in single *trb* (left) and double *trb* mutants (right) compared to Col-0. **D)** Direction of differential expression for comparisons as in **C. E)** comparing DEGs in triple *trb123* mutants against all DEGs detected against Col-0 and between double and single mutants. Significance of overlap in C) and E) determined by SuperExact test using all expressed genes as background. **F)** Heatmap of clusters of all DEGs detected in single and double mutants as in **E)**. Clustering was performed after Variance Stabilization Transformation (VST) of read count data and normalization by the mean for each gene. Sample *trb2.1* was an outlier to all other samples and was excluded from all statistical analysis although later added to the heat-map. **G)** Gene-Ontology (GO) and pathway enrichment analysis for five clusters shown in **F)**. Significance was tested against the background of all expressed genes. Relative enrichment of GO terms for Biological processes (BP), cellular compartment (CC), molecular function (MF) and KEGG

metabolic pathways and wiki pathways (WP) are indicated by bubble sizes, statistical significance by color codes. **H)** Venn diagram showing the number of genes associated with ChIP-seq peaks identified for TRB1, TRB2 and TRB3. Statistical test as in C using all annotated genes as background. **I)** Distribution of TRB1, TRB2 and TRB3 target genes among the transcriptional clusters shown in **F)**. Bar plots show Odds ratio and -log10 pValues as determined by Fisher's Exact test are indicated as color code.

and *TRB3* cannot fully compensate for the loss of *TRB1*. Furthermore, *trb1* mutants showed a clear bias towards genes with decreased expression (66% down vs Col-0), while *trb1 trb3* and *trb2 trb3* were biased towards increased expression (65% and 80% up DEGs vs Col-0, respectively) (Fig 1D). Although only a small number of DEGs were identified in *trb2* and *trb3* single mutants, 473 additional DEGs could be identified when double mutants were compared to the corresponding single mutants. For these additional DEGs an opposing trend between single and double mutants could be observed (S1A-S1C Fig and S1 File).

Only around a third (28%) of the DEGs identified in all single and double mutant comparisons were shared with *trb123* mutants, which represent 12% of the 2634 DEGs identified in these mutants (Fig 1E). While a higher number of DEGs is expected from the strong phenotypic changes observed in *trb123* mutants, the occurrence of genes specific to the milder mutants could be explained by differences in tissue composition.

To gain more insight on the specific impact of *TRB* paralogs on the transcriptome, we performed partitioning around medoid (PAM)-clustering of all DEGs based on their relative expression profile. The number of clusters k = 5 was empirically determined as best (Fig 1F). Cluster 1 showed decreased expression in *trb1* compared to Col-0, but increased expression in *trb1 trb2*. GO-term enrichment analysis found terms related to immune-responses as enriched (Fig 1G). Cluster 2 showed increased expression in *trb1* single mutants, which was further enhanced in *trb1 trb2* and *trb1 trb3* double mutants. Cluster 3 and 4 showed a somewhat reversed pattern, with cluster 3 containing more genes with decreased expression in *trb1-2* compared to Col-0, which was more obvious in *trb1 trb2* and *trb1 trb3* mutants. Cluster 4 was very similar to cluster 3 in *trb1 trb2* and *trb1 trb3* double mutants but not affected in *trb1* single mutants. GO-term enrichment analysis did not associate terms with cluster 2 and 3; however, KEGG terms related to ribosome biogenesis and glucogenesis were enriched, respectively (Fig 1G and S1 File). Cluster 4 enriched GO-terms related to the formation of heterochromatin.

Last, expression in cluster 5 was not altered in single mutants compared to Col-0 but increased in all double mutants, most strongly so in *trb2 trb3*. Genes associated with a response to hypoxia were enriched in cluster 5. Furthermore, cluster 5 enriched the wiki-pathway flowering initiation and flowering development (https://www.wikipathways.org/) based on the presence of the floral organ identity genes *AGAMOUS* (*AG*), *SEPELLATA* (*SEP*) *1* and *3*, the flowering time regulator *TEMPRANILLO 1* (*TEM1*) and the PcG component *EMBRYONIC FLOWER 1* (*EMF1*), which were previously identified as TRB-dependent targets of PRC2 [14] (S2 Fig).

## Binding sites of TRB1-3 reveal distinct target site preferences

To better understand the redundant and specific roles of TRB paralogs, genome-wide binding profiles of all three were compared. TRB ChIP-seq libraries were prepared using chromatin from 14-day old seedlings stably expressing *TRB2* and *TRB3* carboxy-terminal fusions to *YELLOW FLUORESCENT PROTEIN* (*YFP*) under the control of their respective promoters (*TRB2-YFP* in Col-0 background and *TRB3-YFP* in *trb3–2* background, [14]). TRB1-GFP ChIP-seq data were taken from Zhou *et al.* (2016). Enriched peaks were called against control ChIP-seq libraries prepared from Col-0 chromatin precipitated with the same anti-GFP antibody from two biological replicates using the Irreproducible Discovery Correction (IDR) framework at a cut-off of IDR ≤ 0.05 (S3 Fig, [24]). The analysis identified 7483, 3771, 845 peaks for TRB1, TRB2, TRB3 respectively (S4A Fig and S2 File). The observed proportion of peaks for TRB paralogs is similar to those obtained in a previous study by Wang *et al.* [23] in transgenic lines expressing FLAG epitope tagged genomic TRB1, TRB2 and TRB3 fusions. Overall, this study discovered more peaks for all TRBs; however, the relative proportion of peaks

and overlapping peaks was similar (S4B-S4C Fig). Visual inspection of coverage tracks showed that many sites unique to TRB1 showed clearly distinct peaks, while sites unique to defined as TRB2 or TRB3 frequently showed low enrichment (below the significance threshold) of the other TRBs (S4D Fig). We annotated all TRB peaks to target genes to estimate whether transcriptional patterns detected in the double and triple mutants could be related to the binding of TRB paralogs (Fig 1I and S3 File). Overall, DEGs were slightly more likely to be TRB target genes than expected by chance (Fisher's Test, p < 0.010, odds ratio 1.18). We then tested specific TRB paralog enrichment between the five PAM-clusters detected for DEGs (Fig 1F). Cluster 1 showed decreased expression in *trb1* single mutants but contained significantly less direct TRB1 targets than expected (Fig 1I). In contrast, genes in cluster 2 were more likely TRB1–3 targets than expected by chance. Cluster 2 genes show increased expression in *trb1* single and *trb1* containing double mutants, indicating that TRB1 in combination with other TRBs plays a direct repressive role at these genes. A direct repressive role is also supported in cluster 5, which shows increased expression in double mutants, particularly in trb2 trb3. The cluster is significantly enriched for TRB2 and TRB3 target genes (Fig 1F and 1I). Clusters 3 and 4 show a similar pattern with decreased expression in particular in *trb1 trb2* double mutants; however, only cluster 4 is enriched for direct TRB1 and TRB2 target genes, while cluster 3 is even slightly depleted for TRB2 targets. The clusters are differentiated by a decreased expression of cluster 3 in *trb1* single mutants that is not observed in cluster 4. For cluster 4, a direct transcription activation role for TRB1 and TRB2 targets is supported.

Overall, a direct repressive role is supported for the combination of all TRBs and a combination of TRB2 and TRB3. A direct transcription promoting role is supported by the combination of TRB1 and TRB2.

## TRB paralogs show preferred association with distinct chromatin regulatory complexes

To better understand the redundant and specific roles of TRB paralogs within the context of associated chromatin regulatory complexes, we compared the binding of TRB1-3 with available data for TRB-interacting protein complexes. Since there was a considerable difference in the number of targets between our data and the published data for Flag-tagged TRB1-3 from Wang et al. [23], all analyses were performed using both target sets independently. Representative of the PEAT complex, genome-wide binding data were available for core-components EPCR1 and PWWP1 [12] and for UPB5, which removes the PRC1-associated mono-ubiquitination of H2A [13] and was recently proposed to be a PEAT component [11]. HAM1 and EPL1B are members of the NuA4-complex, the former but not the latter was also shown to interact with PEAT [11]. CLF and SWN are core components of the PRC2 complex [21]. The H3K4me3 de-methylase JMJ14 was shown to interact with TRB with a role in balancing gene expression [23]. Since data were generated by different groups and analyzed through different bioinformatic pipelines, we classified peaks into decile bins based on the published significance scores for each protein. We divided the genome of *Arabidopsis thaliana* into 402363 bins of 300 bp length and considered proteins to co-localize if any part of their associated peaks was found in the same bin (S3 File). Pearson's correlation coefficient analysis identified four complex groups with strong correlations within the groups, corresponding to PEAT (EWPCR1, PWWP1, UBI5), NuA4 (HAM, EPL1B), TRBs-JMJ14 (TRB1, TRB2, TRB3, JMJ14) and PRC2 (CLF, SWN) (Fig 2A). Positive correlation was also observed between PEAT, NuA4 and all TRBs in the order of TRB1 > TRB2 > TRB3. For PRC2, correlations were positive for TRB3 and TRB2, while negative for TRB1. In contrast, JMJ14 showed negative correlation with PEAT, NuA4 and PRC2 binding sites, indicating that although JMJ14 and TRBs often bind targets together, they do not usually also bind at PEAT, NuA4 and even PRC2 associated sites. To confirm the correlations observed with our TRB dataset we repeated the correlation analysis with the published dataset for TRBs from Wang et al. [23]. Despite the limited overlap between the peak sets from both studies, the correlations between different TRBs and their binding partners were confirmed indicating that the underlying trends are robust event if target lists change due to experimental conditions (S5A Fig).

A heatmap using the matrix of decile transformed binding sites across target categories underscored that TRBs rarely bind chromatin without any of the described partner complexes (Figs 2B and S5B). The heatmap visualized the mostly

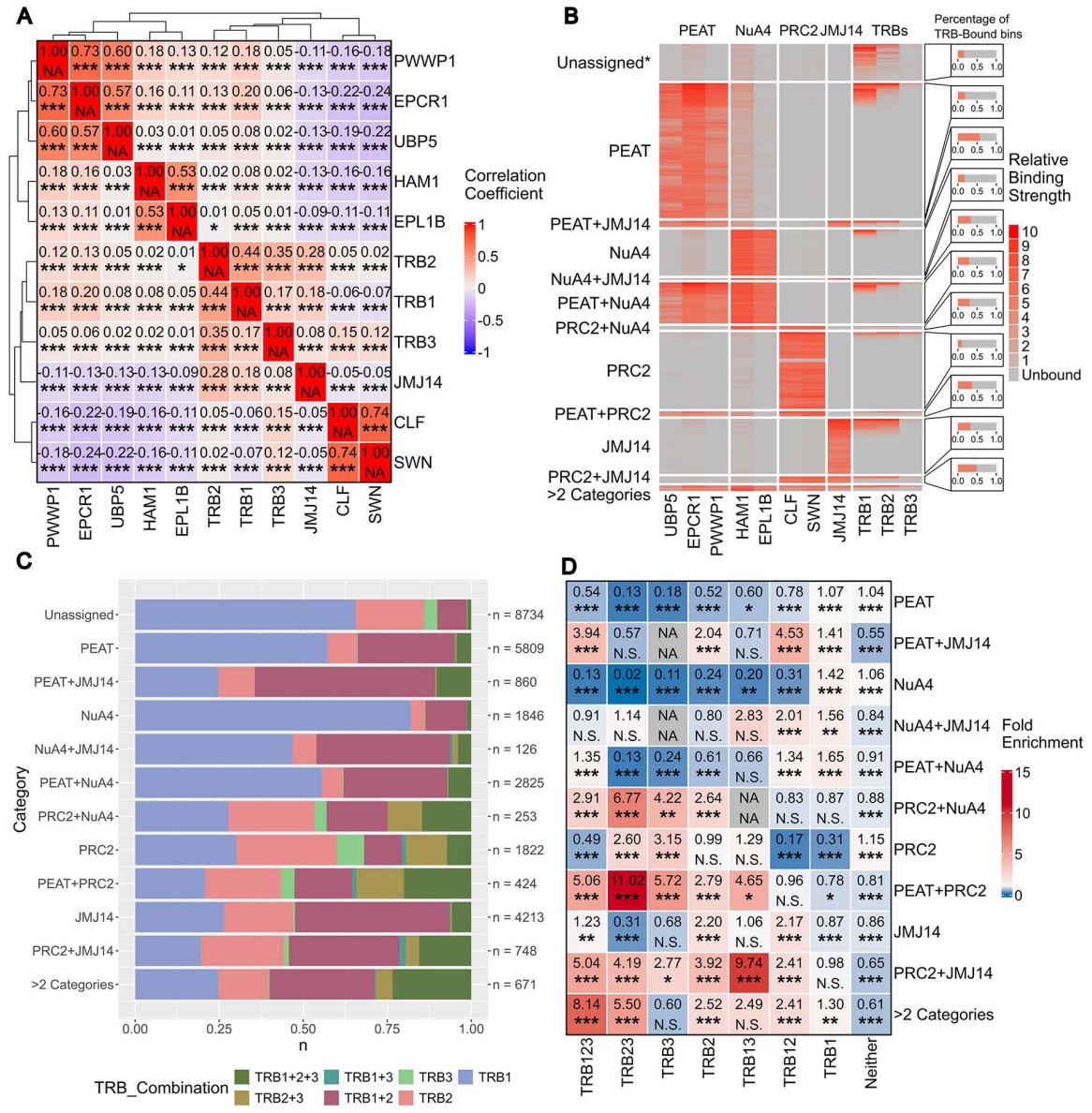

**Fig 2. Analysis of genomic binding locations of eleven epigenetic regulatory proteins. A)** Pairwise Pearson correlation matrix of ChIP-Seq peaks derived from eleven regulatory proteins and assigned to 500 bp genomic bins. Significance levels: ***, p<=0.0005, **, p<=0.005, p<=0.05, N.S., p>0.05. **B)** Left, Heatmap depicting the genomic bins bound by each of the ChIP-Seq sets used in **A**. Relative binding strength of each peak expressed through deciles. Columns depict proteins grouped by regulatory complex and rows depict bins assigned to complexes based on presence of at least two complex components. Bins assigned to more than two complexes were grouped into one category. *, bins which were neither assigned to complexes nor TRB-bound were excluded for the sake of readability. Right, Percentage of TRB-bound bins for each category. **C)** Distribution of TRBs in the TRB-bound bins of each category assigned in **B**. **D)** Pairwise exact test statistics as described by Wang et al. [23] for bins assigned to complex categories and their corresponding TRB combinations. Significance levels: ***, p<=0.0001, **, p<=0.001, p<=0.01, N.S., p>0.01.

exclusive, non-overlapping nature of complex categories except for PEAT and NuA4, which are equally likely to bind chromatin in combination or alone. Next, we evaluated if TRB paralogs showed preference for specific partner complexes. Association of fully resolved TRB binding categories indicated a high proportion of PEAT and NuA4 sites that were

exclusively bound by TRB1 (Figs 2C and S5C). If JMJ14 was also associated with PEAT and/or NuA4, the proportion shifted to include more peaks co-bound by TRB1 and 2 or by all TRBs. The shift in distribution was significantly different from the expected based on a SuperExact Test of expected combinations (Figs 2D and S5D) [25]. Presence of TRB2, in particular in combination with TRB1, was most significantly enriched for JMJ14 containing bins, irrespective of the presence or absence of other interacting complexes. In contrast, PRC2 was most overrepresented at peaks only bound by TRB3, followed by the combination of TRB3 and TRB2. At the rare sites where PRC2 co-bound with JMJ14 or with PEAT, the overrepresentation of TRB1/TRB3 and TRB2/TRB3 peaks became much higher.

Since TRB1 and TRB3 showed a strong propensity to associate respectively with PEAT/NuA4 and PRC2, we performed Immunoprecipitation Mass Spectrometry (IP-MS) to evaluate whether preferred complex associations could also be observed at the protein level. Plants expressing *TRB1* or *TRB3* to *YFP* fusions under the control of the *Cauliflower Mosaic Virus 35S* promoter (*35Sp*) were immunoprecipitated from nuclear extracts using a GFP-trap resin. Samples prepared from *35Sp-ENHANCED DISEASE SUSCEPTIBILITY 1 (EDS1)-YFP* transgenic lines were used as background control. Using biological triplicates, overall, 1149 proteins were identified, of which 96 were significantly enriched in either TRB1 or TRB3 pull-downs or both (S4 File). Co-purified proteins included components of PEAT, NuA4, JMJ14 complexes and PRC2. Only 6 and 7 proteins were significantly enriched in TRB1 over TRB3 and TRB3 over TRB1, respectively (Fig 3A). PEAT-components EPCR1 and EPCR2 showed significant preference to purify with TRB1 over TRB3 while other PEAT components showed a trend towards a higher enrichment for TRB1 (Fig 3B). In contrast, the only NuA4 component that was enriched, TRA1A, was equally enriched by both TRBs (Fig 3C). Equal purification was also observed for JMJ14 components, which showed only a slight bias towards TRB3 (Fig 3D). EMF1 was the only PRC2 associated protein identified and it was exclusively enriched by TRB3 (Fig 3E). Both, TRB1 and TRB3 co-enriched the transcription factor (TF) VIVIPAROUS1/ABI3-LIKE1;VP1/ABI3-LIKE 1 (VAL1), which can recruit PRC1 and PRC2 to target regions.

## TRB1–3 peaks enrich distinct DNA motifs dependent on co-bound regulatory complexes

Analysis of DNA motif enrichment at TRB1-3 peaks classified by co-associated complexes provided further evidence of specialization among the three TRB paralogs. We used the XStreme pipeline of the MEME-Suite to evaluate DNA motif enrichment at TRB1, 2 and 3 peaks as well as the peaks assigned to the different complex categories for each paralog (Fig 4A). A comparison of enriched motifs of all TRB1-3 peaks regardless of complex association revealed that the main telo-box motifs bound by TRB1-3 are not identical. While all three paralogs enrich the canonical "AAACCCT" telo-box motif, TRB1 also significantly enriched the similar but distinct "CRACCTA" motif. While multiple secondary motifs were found for all three paralogs, it stands out that motifs associated with the basic/helix-loop-helix (bHLH) transcription factor family are only enriched in TRB1 ChIP-Seq peaks. Motifs related to TCP-TFs, on the other hand, only enriched in the peaks of TRB1 and TRB2 (Fig 4A).

To identify motifs associating TRBs with one of the four regulatory complexes, we repeated the XStreme analysis pipeline using only TRB peaks from sites that were categorized as exclusive binding locations of PEAT, NuA4, PRC2, or JMJ14. This detailed analysis showed that TRB1 enriches a second telo-box-like motif ("RMCCTA") at sites not co-bound by PRC2. It furthermore indicates that the BPC-transcription-factor-related GAGA-box and NAC-TF associated motifs are universally enriched at all sites, independent of co-bound complex components. Other secondary motifs are specifically enriched in sites co-bound by certain regulatory complexes. This includes the site II motif, which does not enrich at sites with PRC2 binding, ATATAT, which does not enrich at sites assigned to NuA4, and the contrasting bHLH-related motifs, which exclusively enrich at sites with NuA4 binding. Looking closer at the ranks associated with each enriched motif reveals interesting differences between sites with otherwise similar enrichment patterns. Most strikingly, peaks associated with sites of all three paralogs co-bound by JMJ14 show NAC-related TFs as their highest-ranking motif, even surpassing the ranks of telo-box motifs.

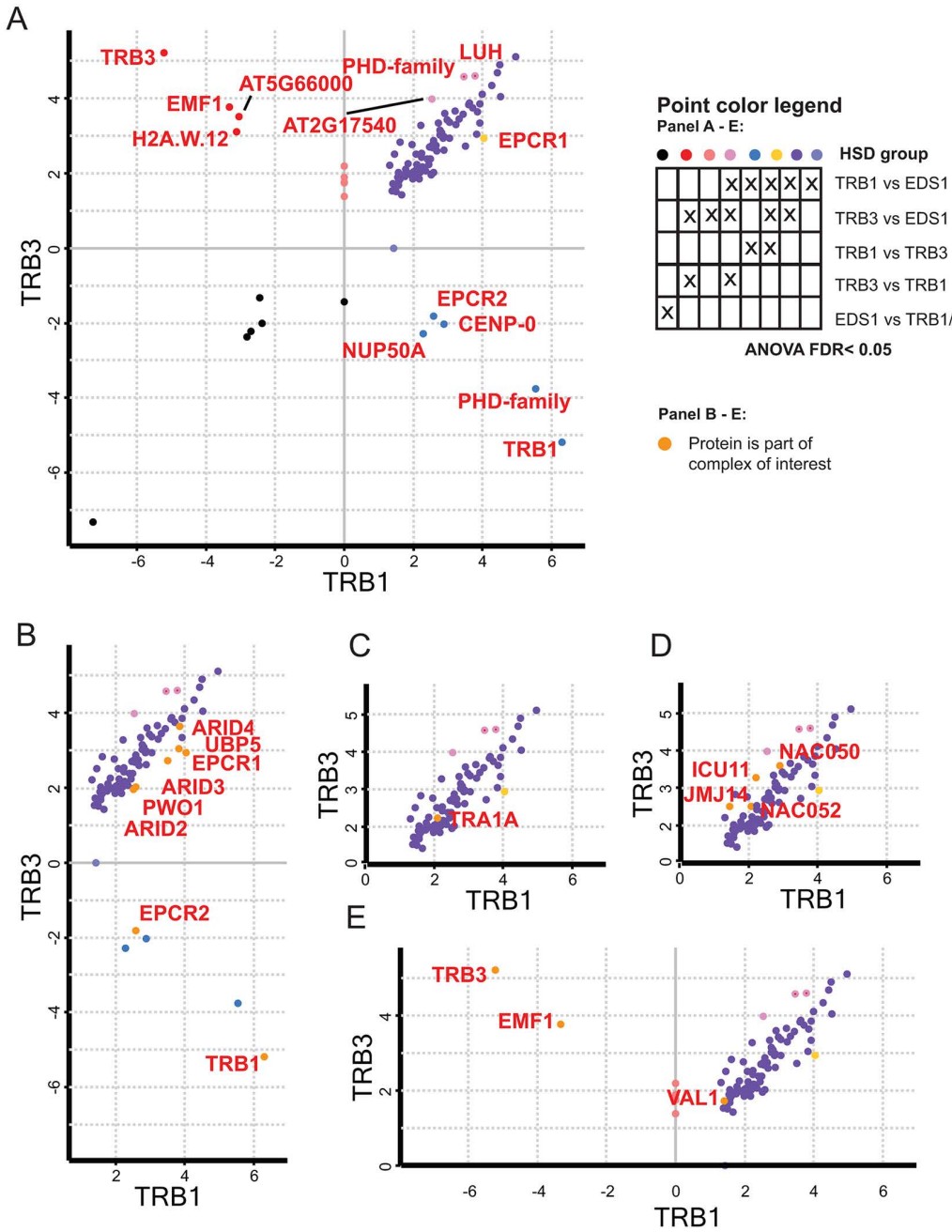

**Fig 3. TRB1 and TRB3 interactome. A)** Comparisons of IP-MS between TRB1 and TRB3 co-purifying proteins in nuclear extracts of 14-day-old seed-lings. Axes show log2-fold enrichment data of log transformed LFQ data from TRB1-GFP vs EDS1-GFP and TRB3-GFP vs EDS1-GFP as indicated). Values with statistical significance (FDR < 0.05) enrichment or depletion as determined by ANOVA and HSD test are indicated. Significance groups indicated by colors as indicated in the legend. Gene symbols are indicated for genes that show significant difference between TRB3 and TRB1. **B)** Excerpt of A) showing PEAT complex components as indicated by color and gene symbol. **C)** excerpt of A) for NuA4 complex. **D)** excerpt of A) for JMJ14 complex. **E)** as in A for PRC complex.

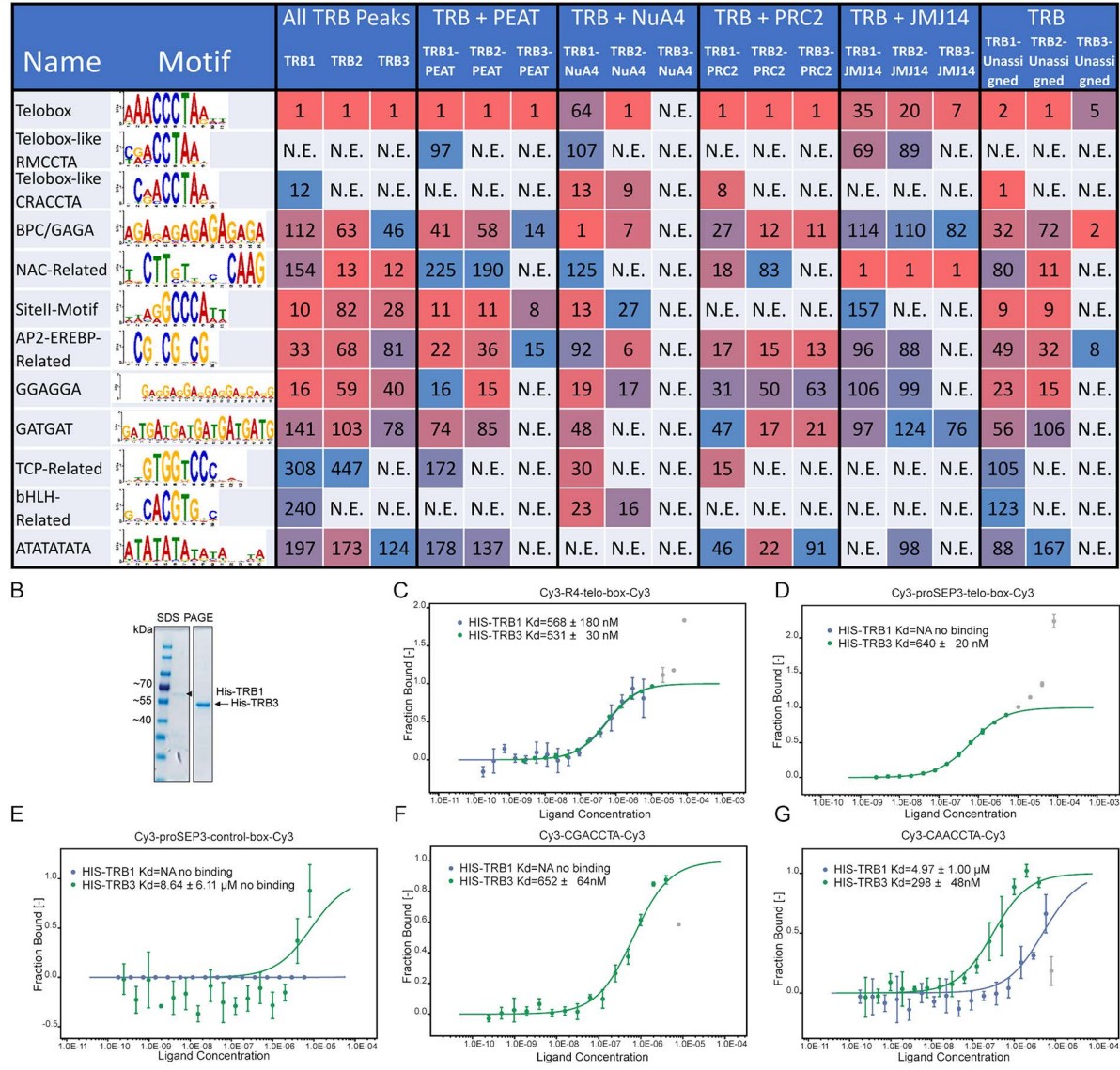

| Name | Motif | All TRB Peaks | | | TRB + PEAT | | | TRB + NuA4 | | | TRB + PRC2 | | | TRB + JMJ14 | | | TRB | | |
|---|---|---|---|---|---|---|---|---|---|---|---|---|---|---|---|---|---|---|---|
| | | TRB1 | TRB2 | TRB3 | TRB1-PEAT | TRB2-PEAT | TRB3-PEAT | TRB1-NuA4 | TRB2-NuA4 | TRB3-NuA4 | TRB1-PRC2 | TRB2-PRC2 | TRB3-PRC2 | TRB1-JMJ14 | TRB2-JMJ14 | TRB3-JMJ14 | TRB1-Unassigned | TRB2-Unassigned | TRB3-Unassigned |
| Telobox | | 1 | 1 | 1 | 1 | 1 | 1 | 64 | 1 | N.E. | 1 | 1 | 1 | 35 | 20 | 7 | 2 | 1 | 5 |
| Telobox-like RMCCTA | | N.E. | N.E. | N.E. | 97 | N.E. | N.E. | 107 | N.E. | N.E. | N.E. | N.E. | N.E. | 69 | 89 | N.E. | N.E. | N.E. | N.E. |
| Telobox-like CRACCTA | | 12 | N.E. | N.E. | N.E. | N.E. | N.E. | 13 | 9 | N.E. | 8 | N.E. | N.E. | N.E. | N.E. | N.E. | 1 | N.E. | N.E. |
| BPC/GAGA | | 112 | 63 | 46 | 41 | 58 | 14 | 1 | 7 | N.E. | 27 | 12 | 11 | 114 | 110 | 82 | 32 | 72 | 2 |
| NAC-Related | | 154 | 13 | 12 | 225 | 190 | N.E. | 125 | N.E. | N.E. | 18 | 83 | N.E. | 1 | 1 | 1 | 80 | 11 | N.E. |
| SiteII-Motif | | 10 | 82 | 28 | 11 | 11 | 8 | 13 | 27 | N.E. | N.E. | N.E. | N.E. | 157 | N.E. | N.E. | 9 | 9 | N.E. |
| AP2-EREBP-Related | | 33 | 68 | 81 | 22 | 36 | 15 | 92 | 6 | N.E. | 17 | 15 | 13 | 96 | 88 | N.E. | 49 | 32 | 8 |
| GGAGGA | | 16 | 59 | 40 | 16 | 15 | N.E. | 19 | 17 | N.E. | 31 | 50 | 63 | 106 | 99 | N.E. | 23 | 15 | N.E. |
| GATGAT | | 141 | 103 | 78 | 74 | 85 | N.E. | 48 | N.E. | N.E. | 47 | 17 | 21 | 97 | 124 | 76 | 56 | 106 | N.E. |
| TCP-Related | | 308 | 447 | N.E. | 172 | N.E. | N.E. | 30 | N.E. | N.E. | 15 | N.E. | N.E. | N.E. | N.E. | N.E. | 105 | N.E. | N.E. |
| bHLH-Related | | 240 | N.E. | N.E. | N.E. | N.E. | N.E. | 23 | 16 | N.E. | N.E. | N.E. | N.E. | N.E. | N.E. | N.E. | 123 | N.E. | N.E. |
| ATATATATA | | 197 | 173 | 124 | 178 | 137 | N.E. | N.E. | N.E. | N.E. | 46 | 22 | 91 | N.E. | 98 | N.E. | 88 | 167 | N.E. |

**Fig 4. Analysis of TRB bound and associated motifs. A)** DNA motifs enriched in ChIP-Seq peaks of TRB1, TRB2, and TRB3. Either all peaks, peaks assigned to gene regulatory complex binding sites or only peaks without complex assignment were used for the analysis as indicated in the header. Numbers indicate enrichment rank in the MEME-Suite results. **B)** Bacterial expressed HIS-TRB1 and HIS-TRB3 after affinity purification and SDS-PAGE. **C)** Fraction of Cy3-labelled double stranded 4x repeated telo-box probe bound by HIS-TRB1 and HIS-TRB3. **D)** as C) using Cy3 lablelled telo-box motif as found in the promoter of SEP3. **E)** as C) using a Cy3-labelled control region from the SEP3 promoter **F)** and **G)** as C using Cy3 labelled double stranded oligos containing the CGACCTA and CAACCTA motif, respectively.

These results indicate a substantial difference in DNA motifs found at sites where TRBs bind together with PEAT, NuA4, PRC2, or JMJ14 and could therefore indicate the role of different TFs in determining the targeting of epigenetic regulatory complexes.

## TRB1 binding to non-canonical telo-box motifs is not explained by affinity to single motifs

Since differences in the telo-box related consensus motifs suggested that the DNA binding properties may differ between TRB1 and TRB2/3, we tested the affinity of TRB1 and TRB3 to single telo-box and the two variants of the "CRACCTA"

motifs by microscale thermophoresis (MST). MST determines dissociation constants between fluorescently labelled targets and unlabelled ligands by measuring changes in the velocity by which the fluorophore moves in or out of 1–6 K temperature gradients (Seidel et al. 2013) [26]. TRB1 and TRB3 were previously shown to bind probes containing four telo-boxes (R4) that imitate telomeric repeats *in vitro* [27].

We could reproduce binding to R4 probes for bacterially expressed six histidine (HIS)-tagged TRB1 and TRB3 with binding affinities that were in the range of those previously reported ($K_D = 567 \pm 180$nM and $K_D = 531 \pm 30$nM for TRB1 and TRB3, respectively) (Fig 4B and 4C). In contrast, only HIS-TRB3 interacted with an 28 bp oligomer derived from the *SEP3* promoter containing a single telo-box ($K_D = 640 \pm 19$ nM) (Fig 4D and 4E). We used the telo-box of the *SEP3* promoter since we had previously established that the motif is critical to recruit PRC2 to the *SEP3* locus [14]. Since the affinity of HIS-TRB3 towards the single telo-box motif was comparable to that of the R4 telo-box repeat, a non-cooperative interaction mechanism between TRB3 and DNA is probable. Motif enrichment suggested that TRB1 may bind to single CRACCTA motifs; however, HIS-TRB1 did not bind to the CGACCTA variant and only with low affinity to the CAACCTA variant ($K_D = 4.9 \pm 1.0$ µM) (Fig 4F and 4G). Unexpectedly, HIS-TRB3 bound CGACCTA with similar affinity as single telo-boxes (CGACCTAA: $K_D = 652 \pm 64$ nM) and CAACCTA slightly better ($K_D = 298 \pm 48$ nM) (Fig 4F and 4G).

In sum, TRB1 seems unable to bind single telo-box and CRAACTA motifs *in vitro,* indicating that *in vivo* enrichment of these motifs by TRB1 may depend on the presence of other factors.

## Genes bound by different regulatory complexes participate in distinct biological processes, dependent on co-bound TRBs

Next, we investigated whether the epigenetic regulatory complexes diverged not only in their binding sites and TRB paralog association, but whether their target genes were also involved in different biological processes. Through the annotatePeak function of the ChIPseeker library, peaks were annotated to the *A. thaliana* genome. Peaks covering multiple genes were annotated to all overlapping genes. This revealed that, while only 8.6% of the 300-bp bins were bound by TRBs, 36.2% of all annotated nuclear genes exhibited binding of at least one TRB.

We started by identifying the most enriched GO-terms for each complex, regardless of the TRB paralog they were associated with (Fig 5A and S5 File). Overall, it was possible to assign specific functions to each complex combination. Ribosome-related GO-terms were significantly enriched in genes assigned to PEAT with and without NuA4; however, the significance was higher for PEAT only genes, indicating a previously undescribed role of the PEAT complex in ribosomal regulation. PRC2 assigned genes enriched terms related to plant development in general and floral development in particular. This is consistent with previously published functions of TRB-PRC2 complexes [14]. NuA4 in combination with PRC2 enriched terms associated with regulation of both chlorophyll and DNA/RNA synthesis. The combination of JMJ14 and NuA4 target genes on the other hand, enriched GO-terms related to starvation responses. Last, the five GO-terms with most enrichment in the unassigned set of genes were all related to oxygen response/hypoxia, indicating that our current understanding of TRB-related gene regulation is still incomplete.

Since epigenetic regulatory complexes can enriched different GO-terms, based on which TRB paralog combination they associated with, we took a closer look at the GO-terms and split the gene sets based on co-bound TRB paralogs. (Figs 5B-5D and S6). For further analysis we focused on the genes assigned to only PEAT, PRC2, or NuA4 and JMJ14, as these combinations enriched the GO-Terms that were of most interest to us. Comparing the TRB-paralog combinations, it became clear that the affinity of PEAT towards ribosomal and RNA-processing genes is driven largely by TRB1, as almost all possible TRB1-containing combinations enrich these terms (Figs 5B and S6B). Interestingly, genes bound by PEAT and only TRB1 or TRB2 additionally enriched terms related to chromatin organization. PRC2 target genes involved in floral development are only enriched in the genes that are also bound by TRB2 and TRB3 or all three TRB paralogs (Fig 5C). The role of NuA4 in combination with JMJ14 in regulating catabolic processes and ion transport appears to be primarily associated with sites that are bound by TRB1 and TRB2 (Fig 5D). While genes assigned to only NuA4 alone did

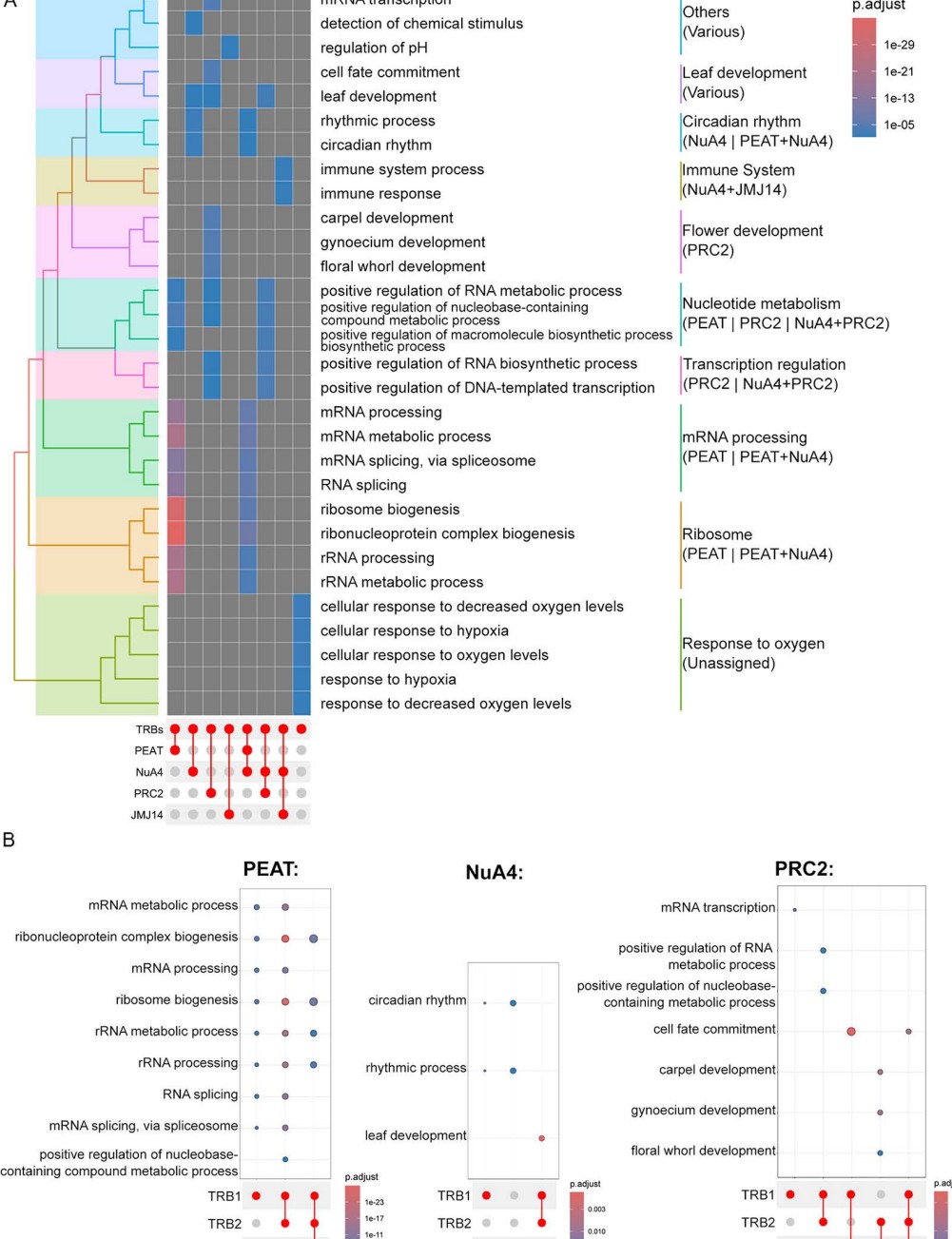

**Fig 5. Gene ontology enrichment analysis. A)** Enriched GO-Terms for genes assigned to either no complex (Unassigned), PEAT, NuA4, PRC2, JMJ14, or combinations of up to two complexes. Only the 5 most enriched GO-Terms of the "Biological Process" aspect are shown. Similar terms are summarized at the bottom. **B-D)** Enriched GO-Terms for the gene sets assigned exclusively to PEAT **(B)**, PRC2 **(C)**, or NuA4+JMJ14 **(D)**. Sets were separated by co-bound TRB-paralogs. The 5 most enriched "Biological Process" terms per combination were selected and similar terms reduced into one representative term. Full list of enriched GO-terms available in S5 File.

not enrich any GO-terms, exclusively JMJ14 assigned genes did enrich terms associated with actin organization, membrane transport, and seed growth regulation (S6D Fig).

## Transcriptomic changes in *trb* mutants are partially explained by the assigned regulatory complexes

In order to validate the complex assignments and to improve our understanding of the effects of TRBs on gene expression, we tested whether specific complexes are over- or underrepresented in the sets of genes assigned to the 5 PAM-clusters identified for DEGs in single and double *trb* mutants (Fig 1F). The normalized average expression profile across all genes per cluster recalls the expression trends of each cluster (Fig 6A). Using the gene-to-complex assignments produced for the gene ontology analysis (Supplemental data 3), we intersected the PAM-clusters with complex combinations and performed multi-set interaction analysis according to [25] (Fig 6B). Starting with clusters 2 and 5, which indicated a repressive role for TRBs and had been detected as overrepresented for target genes of all TRBs and TRB2/TRB3, respectively (Fig 1I). Cluster 2 was overrepresented for the combination TRB with PRC2 (1.95-fold over expected), with PRC2 and JMJ14 (2.19-fold) but also PRC2 and PEAT (1.89-fold). Overall, the enrichment of repressive chromatin complexes may explain the upregulation of target genes in *trb* mutants. Cluster 5 was enriched for genes bound by TRBs in combination with PRC2 and PEAT (3.38-fold) while depleted for TRB and only PEAT complexes (0.16-fold). Loss of PRC2 recruitment in combination with maintained PEAT recruitment could explain the high induction of expression in *trb2 trb3* mutants (Fig 1I).

Cluster 4 showed decreased expression in particular in *trb1 trb2* mutants and was enriched for direct TRB1 and TRB2 target genes (Figs 1F, 1I and 6A). The enrichment of TRB and PEAT targets (1.65-fold) combined with the depletion of TRB and JMJ14 targets (0.20-fold) supports a direct activating role for TRB1 and TRB2 in this cluster. Although cluster 3 was not enriched for TRB target genes, it showed similar complex association as cluster4 (Fig 6B). Genes in cluster 1 were less likely to be direct target genes than expected, but if those that were targets, were likely to also associate with both activating NuA4 and repressive JMJ14 complexes (5.34-fold). Cluster1's response to mutants is inconsistent, which could be explained by the targeting of complexes with opposing function.

In sum, the combined analysis of transcriptome changes and complex target group categories illustrates that the effect of TRBs on expression is strongly dependent on their associated partners. In particular in cases when genes are bound by several TRB associated complexes with opposing effects, TRBs could be involved in maintaining a balance between repression and active transcription.

## Discussion

### TRBs associate with distinct epigenetic regulatory complexes in a primarily mutually exclusive manner

Since it was first discovered that TRBs can participate in multiple epigenetic regulatory complexes, it had been unclear whether TRB-bound genomic sites act as "landing sites" for multiple complexes, or whether each site is associated with only a single complex. Our analysis of the ChIP-Seq-determined binding sites of eleven complex-forming proteins (including two novel TRB sets) serves as a useful tool to evaluate the overall binding behavior of TRBs. According to our analysis, it is likely that the majority of TRB-binding sites are associated with a single regulatory complex (Figs 2 and S5). The only exception seems to be PEAT and NuA4, which share a significant proportion of their binding sites, likely caused by the role of HAM1 in both complexes [12,28]. It should, however, be noted that all the ChIP-Seq datasets included in this study are obtained from whole seedlings. It is therefore likely that temporal and/or tissue-specific binding differences are obscured in our analysis. In addition to mutual exclusivity, we also discovered that a large portion of the binding sites of the four complexes included in this study were not bound by any of the three TRB paralogs. In the case of PEAT, this is unexpected, since TRBs have been proposed as core components [12]. In part, this finding may be explained by differences in stringency thresholds defining binding sites in different bioinformatics pipelines; however, the same phenomenon

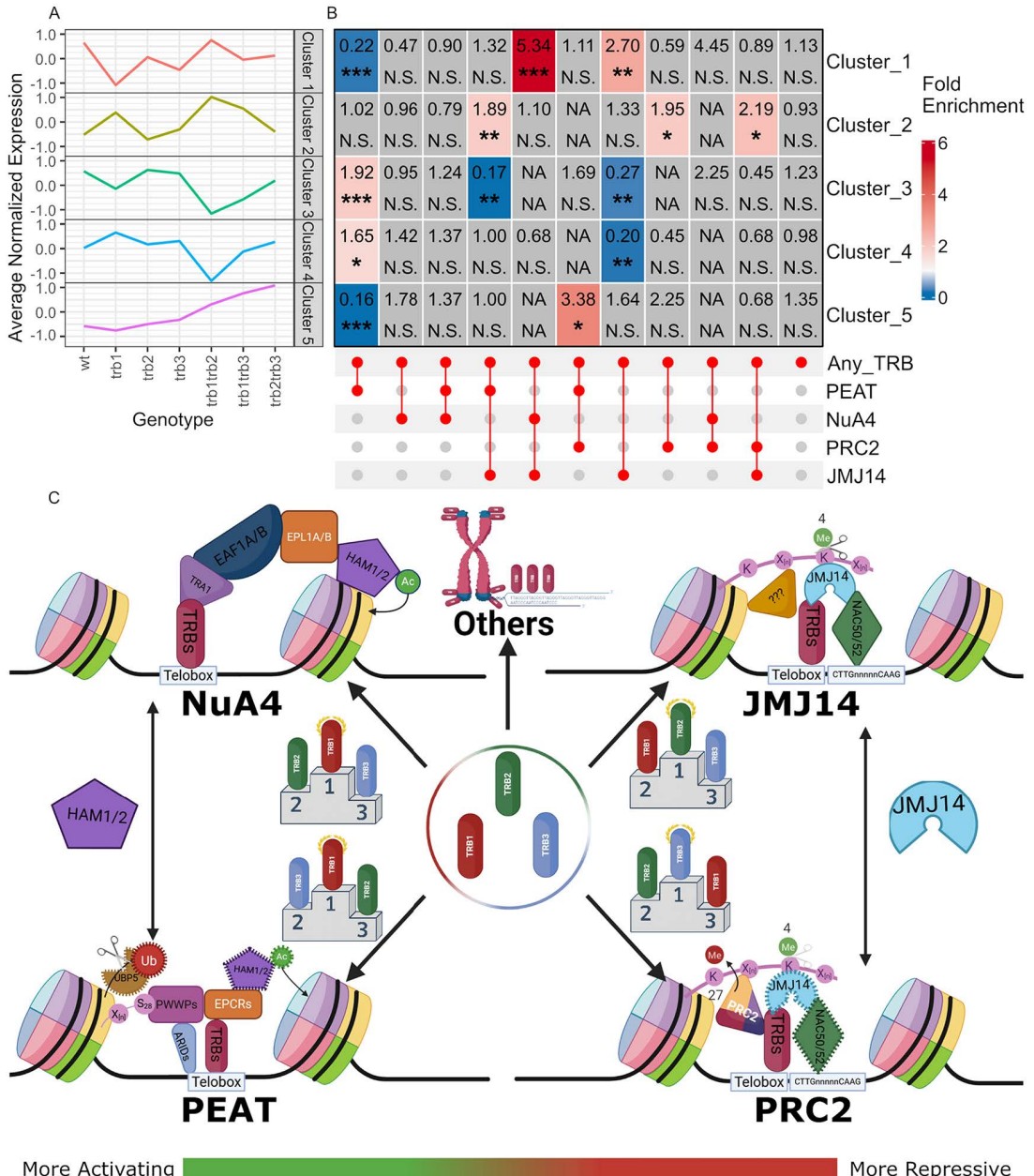

**Fig 6. TRB-mediated gene regulation in epigenetic context. A)** Average normalized expression of the transcription clusters displayed in Fig 1F. Values of three replicates of each of genotypes were averaged. **B)** Pairwise Exact test statistics as described by Wang et al. [25] for the same clusters as A intersected with the sets of genes annotated to either one, two, or none of the epigenetic regulatory complexes. Significance levels: ***, p<=0.0005, **, p<=0.005, p<=0.05, N.S., p>0.01. **C)** Model of TRB-mediated epigenetic gene regulation. Complex components which are not core to the TRB-containing complexes are indicated with dashed outlines. TRB-paralog specialization is conveyed via podiums. "???" indicates potential unknown complex participants. HAM1/2 and JMJ14 are shared between the complexes on the activating and repressive side, respectively. Created in BioRender. Mendler, M. (2026) https://BioRender.com/1npu38m.

was observed on the bases of an independent dataset for TRB binding sites (S5 Fig). An alternative explanation is that TRBs are optional members of chromatin regulatory complexes and display a more dynamic binding pattern as anticipated. Future studies are requited to clarify this point.

Our analysis provides strong evidence of additional, thus far undescribed, TRB-containing complexes. First, the presence of a large contingency of TRB-JMJ14 co-binding sites in the absence of other complex components strongly indicates the presence of a JMJ14-mediated histone modification complex that acts independently of PRC2, while maintaining JMJ14 association with TRBs. NAC50/52, two strong TRB interactors with slight preference for TRB3 over TRB1, have already been shown to form regulatory complexes with JMJ14 [22]. Furthermore, a recent study has described two novel TRB-associated complex, TRB1/2/3-HELIX-TURN-HELIX-PROTEIN (TRHT) and TRB1/2/3-HISTONE-DEMETHYLASE (TRHD), which are linked to H3K4me3 de-methylation [29].

In addition, the evidence clearly shows a split between PEAT and NuA4 bound TRB-sites, expanding on the previously observed role of TRBs in PEAT complexes which include the HAM1 component of NuA4 [11]. This serves as evidence of the presence of an additional NuA4-TRB complex targeting sites independent of PEAT. In addition to their genomic binding, evidence of these two proposed complexes can also be seen in the IP-MS data. TRA1, which had previously been described as the TF-binding domain of the NuA4 complex [28], is the NuA4 complex component with the strongest TRB1/3 binding score (Fig 3). Last, a substantial number of target sites is associated with TRBs in absence of all other chromatin complex representatives. It is to be expected that other TRB partners will be identified to co-occur at these sites.

### The three paralogs TRB1, 2 and 3 specialize in specific roles, without losing their redundancy

Although reverse genetics suggested almost complete redundancy between the three paralogs TRB1, 2, and 3, whether this redundancy is replicated on a molecular level had never been studied in detail. On the molecular level, TRBs exhibited an unexpectedly strong degree of specialization. In particular, a clear split between TRB1 and TRB2/3 could be detected throughout the analysis. The association between TRB1 and activating complexes such as PEAT and NuA4 could be seen in the overrepresentation of TRB1 bound sites in the ChIP-Seq data of these complexes; furthermore, it is corroborated by GO-term analysis since terms enriched for TRB1 a more significantly overrepresented for TRB-PEAT and TRB-NuA4 target genes (Figs 2 and 5). Conversely, the same data indicate that TRB2/3 associate more strongly with the repressive complex PRC2 and the proposed JMJ14-containing complex. A similar trend can be observed in the IP-MS data comparing TRB1 and TRB3 co-purifying proteins since the only identified PRC2-associated protein, EMF1, was exclusive to TRB3 while PEAT-component EPCR1 was only co-purified with TRB1 (Fig 3).

### Genes that are targeted by more than one TRB-associated complexes are more likely to be differentially expressed in single and double *trb* mutants

Although distinct TRB-associated complexes are rarely found at the same genomic site, genes can be targeted by more than one complex and are likely to show differential expression in *trb* single and double mutants (Fig 6). The implication of several TRB containing complexes with opposing function on transcription indicates a role for TRBs in maintaining the balance between gene activation and repression, at least at a small subset of target genes. The relative preference of TRB1 for activating complexes such as PEAT and TRB2/3 for repressive complexes such as PRC2, could rationalize transcriptional outcomes in the mutants, such as increased expression in *trb2 trb3* double mutants in cluster 5 due to an absence of PRC2 recruitment in the continued presence of PEAT, which is predominantly associated with TRB1. Overall, the effects of incomplete loss of the TRB1-3 clade are mild compared to the effects observed in the triple mutants indicating that an incomplete specialization between TRBs is important for the maintenance of expression homeostasis.

**One allele of TRB2 is not sufficient to maintain normal flowering in long days**

Despite their predominant redundancy, *TRB* alleles are not equal as we found that *prope triple* mutants maintaining only one functional copy of TRB2 flowered early in long day growth condition, while one functional allele of *TRB1* or *TRB3* allowed plants to flower as wild-type controls (Fig 1B). TRB2 is phylogenetically more related to TRB3 than TRB1 [1]; furthermore, we and others found more target genes for TRB2 than for TRB3 (S4 Fig). In their correlation to other chromatin factors, TRB2 is usually positioned between TRB1 and TRB3, which could be interpreted as a less specialized role, but this does not explain why TRB2 represents a potentially weaker allele. The combination of TRB1 and TRB3 is particularly often enriched at PRC2 and JMJ14 bound sites (Figs 2D and S5D). Early flowering has also been reported for *jmj14* mutants and mutants of their interacting TFs, NAC50/NAC52 although the genetic network that underpins this response has not been resolved [22]. Whether the lack of effective recruitment of JMJ14 to target regions by TRB2 explains early flowering will have to be clarified in future studies.

**TRBs are low affinity transcription factors explaining their dependence on other interaction partners and motifs for target association**

Although TRB proteins can bind telo-box and related motifs, their affinity towards tandemly repeated motifs was moderate at best and weak to non-binding in the case of single telo-box or related motifs (Fig 4). Hypothetically, high affinity binding to telo-box motifs would strongly tether TRBs to telomeres, possibly preventing their binding to single motifs at genic regions. On the other hand, the weak affinity of TRBs to target sites *in vitro* suggests that TRBs require assistance to associate with genic target regions *in vivo*. TRBs feature a H1/H5-related domain that may increase the affinity to well-positioned telo-box motifs by interacting with core nucleosomes. Previous studies have shown that presence of the linker histone H1 competes with TRB1 binding to telomere-derived regions [8]. Alternatively, the association with chromatin complexes and other TFs may stabilize TRBs at single telo-box motifs.

While the analysis of DNA motifs could not fully explain likely (co)-recruiting motifs for different TRB-associated complexes, it provided further evidence of specialization. A few DNA motifs show differential enrichment between the regulatory complexes (e.g., the site II motif and bHLH related motifs) and ranks of the enriched motifs differ substantially between the complex target sites (Fig 4). Most obviously, sites of TRB-JMJ14 binding enrich NAC-related motifs to a stronger degree than telo-box motifs. Several studies had already presented interaction data identifying a complex of NAC50/52 with TRBs and JMJ14 [22,23], which was also confirmed in our study (Fig 3). The predominant enrichment of NAC motifs indicates a TRB recruitment mechanism as part of the TRB-NAC-JMJ14 complex involving NAC's affinity to their cognate motifs.

In a second case, interaction between two closely linked motifs appears not to rely on a common complex. Earlier studies revealed that telo-box motifs in combination with site II motifs are able to act as transcriptional enhancers at ribosomal protein encoding genes [30]. Our analysis showed that TRB-PEAT bound genes enrich site II motifs and are associated with GO-terms associated with ribosomal functions (Figs 2 and 4). In contrast, the site II motif binding TEO-SINTE BRANCHED1/CYCLOIDEA/PROLIFERATING CELL FACTOR (TCP) family TFs were not enriched in our IP-MS data (Fig 3).

Last, no singular motif was found that only enriches at sites co-occupied by PRC2 and TRBs. While GAGA-motifs, which are bound by TFs of the BPC family, were previously proposed a co-recruiters of PRC1 or PRC2 components [19,31,32], the corresponding motifs are also enriched at TRB-PEAT and TRB-NuA4 leaving the question of how PRC2-TRB sites are defined unanswered for now. BPC TFs were not enriched in our IP-MS dataset.

In summary, we observed an overall trend of TRB specialization, which can be best described as a preference for specific sites and complex partners (Fig 6C). TRB1 appears to have a strong affinity with PEAT and NuA4, but is also present at many unrelated binding sites. By contrast, TRB3 and TRB2 are predominantly, though not exclusively, found at PRC2

and JMJ14 locations, respectively. These results are consistent with the remarkable redundancy observed at the phenotypic level. It is possible that TRBs are at the beginning of a process of functional specification. Alternatively, their partial functionalization could provide robustness to epigenetic gene regulation, particularly if changes in their relative abundance and activity are connected to environmental or developmental cues. Further studies are required to answer these questions.

## Materials and methods

### Plant material and growth conditions

Plants were grown in greenhouse conditions or growth chambers as indicated under long day (LD) (16h light, 8h dark) photoperiod at 22°C ambient temperature. Plants were randomized within trays for phenotypic analyses. Plants for RNA-seq analysis were grown in growth chambers. Three biological replicates were grown in 1 week intervals in the same chamber, material was collected at ZT10 from 14-day-old seedlings. For ChIP-seq, plants were grown in tissue culture on GM plates in LD conditions at 21°C. Material from replicated plates was collected from 14-day-old-seedling at ZT10 as biological replicates. The *trb1-2* (Salk_001540) and *trb3-2* (Salk_134641) alleles are previously described T-DNA insertion lines, *trb2-2* and *trb2-3* are CRISPR/Cas9 edited alleles as previously described except that the editing transgene was removed by segregation [14]. Double and triple mutants were generated by crosses. Transgenic lines TRB2pro-TRB2-YFP and TRB3pro-TRB3-YFP were previously described [14].

### Scoring of flowering time

Flowering time was scored as the number of leaves at the main shoot (rosette and cauline leaves). Statistical analysis was done by ANOVA with HSD grouping using the agricolae package in R. To distinguish segregating *prope* triple from double mutants, genotyping of individual plants was carried out on genomic DNA prepared using Biospring96 (Qiagen) using manufacturer's instructions. Alleles trb1-2 and trb3-2 were amplified using SALK_LBb1.3: ATTTTGCCGATTTCGGAAC in combination with SALK_001540_RP: ATGCCACCACAATAAATCTCG and SALK_13464_RP: ATGGTTCACGAGAAACCTGTG, respectively. To distinguish between trb2-3 and TRB2, two reactions were carried out for 28 cycles at 62°C annealing temperature dCAPS_TRB2_R: ATTGCCTCAAAGATGATCTTATCC in combination with 8-18-10-specifi: ACTTCCCCCGGAGGTTCTTG and 8-18-10-WT: ACTTCCCCCGGAGGTTCTG, respectively.

### RNA preparation and RNA-seq

Total RNA was extracted from 3-4 14-d-old-seedlings with an RNeasy Plant Mini kit (Qiagen) according to the manufacturer's instructions. To remove gDNA contamination, 10 µg of total RNA was DNase I treated, using the DNA-free DNA Removal kit (Invitrogen), as described in the kit's instructions. RNA quality was assessed by Agarose Gel electrophoreses of an 200ng aliquot DNaseI treated RNA. The RNA samples were sent to BGI TECH SOLUTIONS (HONG-KONG) for poly-A enrichment, library preparation and directional paired-end Nanoball sequencing on the DNBSEQ platform.

### RNA-seq analysis

Paired end reads were mapped to the Arabidopsis thaliana TAIR10 reference genome indexed with the Araport11 genome annotation using STAR. Read counts were pooled for all splice variants as per gene counts. Sense strand gene counts were used for differential expression analysis with the R package DESeq2 using a threshold of padj < 0.05 to set differential expression of mutants vs Col-0 and of double mutants to their respective single mutants. Sample correlation clustering revealed that the *trb2.1* library was as an outlier and was excluded from all statistical analysis. Venn diagrams and statistical testing of overlaps between samples used R packages ggvenn and SuperExactTest, respectively. Clustering of

expression data and drawing of gene-normalized expression heatmaps were carried out using R package ComplexHeatmap using PAM-clustering.

## ChIP and ChIP-seq library preparation

For all ChIP experiments, 2g of 14-d-old seedlings were collected in 50 ml 1x PBS buffer (137mM NaCl, 1.8mM $KH_2PO_4$, 10.1mM $Na_2HPO_4$, 2.7mM KCl), fixed with 1% formaldehyde under vacuum two times for 10 min after which the crosslinking reaction was quenched with 5 ml glycine (1 M) under vacuum for 5 min. Fixed plant material was collected in a sieve, washed with autoclaved water, and dried with paper towels before being snap frozen with liquid $N_2$. Frozen samples were ground at 7200 rpm three times for 30 s, using the Precellys Evolution Homogenizer in combination with a Cryolys Cooling Option (Bertin Instruments) in 7 ml reaction tubes with 3mm ceramic beads.

To extract nuclei, the ground samples were mixed with 30 ml NIB buffer (50mM HEPES-NaOH (pH 7.4), 5mM $MgCl_2$, 25mM NaCl, 5% sucrose, 30% glycerine, 0.25% Triton X 100, freshly add: 0.1% β-mercaptoethanol, 0.1% SIGMA proteinase inhibitor), vortexed, filtrated using Miracloth (Merk) and spun down at 4000rpm and 4 C for 10min. The pellet was resuspended in 20ml 1x Washing buffer (16.7mM HEPES-NaOH (pH 7.4), 6.7mM $MgCl_2$, 33.3mM NaCl, 13.3% sucrose, 13.3% glycerine, 0.25% Triton X 100, freshly add: 0.001% β-mercaptoethanol, 0.001% SIGMA proteinase inhibitor) and spun down at 4000rpm and 4°C for 10min. Then, extracted nuclei were resuspended in TE-SDS (1mM EDTA (pH 8.0), 10mM Tris-HCl (pH 7.4), 0.25% SDS) in a total volume of 600µl, rotated at 4°C and 12 rpm for 20 min, split in 2x300µl and sonicated with a Bioruptor Sonicator (Diagenode) that was attached to a Minichiller cooling system (huber) (Programme: red- 0.5 (on); green - 1 (off); 15 min, H) to produce DNA fragments of ~200–500 bp. Sonicated chromatin was separated from debris by centrifugation at 4°C and maximum speed for 10 min. For ChIP-seq, 400µl sonicated chromatin were mixed with 600 µl of IP dilution buffer (80mM Tris-HCl (pH 7.4), 230mM NaCl, 1.7% NP40, 0.17% DOC), 2µl RNase I (10 mg/ml), 2µl DTT (1M), and 2µl SIGMA proteinase inhibitor. Afterwards, equal volumes of the sonicated chromatin mix were split into two different tubes, 5µl of α-GFP (ab290, Abcam) were added to carry out IP. Samples were rotated at 4°C, 12 rpm overnight in a bohemian wheel. After overnight incubation, unspecific precipitates were removed by centrifugation (4°C, 20000g, 10 min) and the supernatant transferred to a tube containing 30µl rProtein A Sepharose Fast Flow antibody purification resin (GE Healthcare) beads equilibrated in RIPA buffer (0.6x IP Dilution buffer, 0.1% SDS). Samples were rotated at 12rpm and 4°C for 3 h. After centrifugation, 200µl of the supernatant from control samples was reserved as input and kept on ice. Beads were washed with 1 ml RIPA for five times to remove the background. At the 5[th] time, the samples were transferred to fresh tubes with 800µl RIPA and protein-DNA complexes were eluted from precipitated beads by mixing them two times with 160µl glycine elution buffer at RT. IP samples were neutralised with 80 µl of Tris-HCl (1 M, pH9.7). IP samples were de-crosslinked by adding 8µl SDS (10%) and 5µl proteinase K (5mg/ml). For input samples, only 5µl proteinase K was added. DNA was extracted twice with equal amounts of phenol/chloroform and precipitated with 1/10 volumes NaAC (3M), 2.5 volumes EtOH (100%), and 1µl glycogen (10mg/ml) at -20° for 3h. Afterwards, the DNA was washed with 1ml EtOH (70%), dried, and resuspended in 14µl $H_2O$.

For ChIP–seq library preparation, two independent immunoprecipitations for Col-0, TRB2pro-TRB2-YFP and TRB3pro-TRB3-YFP were processed. Libraries were prepared with Ovation Ultralow Library System (NuGEN) according to the manufacturer's instructions, using 71% (10µl) of each ChIP as starting material. Before amplification DNA concentration was measured, using a Qbit 4 (ThermoFisher Scientific), to determine the appropriate number of PCR cycles needed for each sample (see manufacturer's manual). After amplification, DNA was run on a 2% low-melt agarose gel and fragments between 200 and 500 bp length were purified using the MinElute Gel Extraction Kit (Qiagen) according to the manufacturer's instructions except that gel fragments were solved at RT and eluted in 15µl EB buffer. An aliquot of each library was tested via qPCR before and after PCR amplification to confirm that libraries showed similar fold-change between control and target regions. Sequencing was performed as single-end 100-nt reads (ca 13 mio reads/sample) on the Illumina HiSeq3000 platform by the Max Planck Genome Centre Cologne.

## ChIP-seq analysis

After sequencing, adapter sequences ≥ 12 bp were removed using Cutadapt [33]. Reads were aligned to the *A. thaliana* genome (TAIR10) with the Burrow-Wheeler Aligner (BWA) [34] and BAM-files created using SAMtools [35]. SAMtools was used to remove multi-mapping reads by filtering with MAPQ score<10, which resulted in 8.8 to 12.2 million reads per sample. Unique BAM-files were indexed with SAMtools, normalised to Counts Per Million mapped reads (CPM), and converted to bigWig-files using bamCoverage of the deeptools2 suite [36] for visualization in the Integrated Genome Viewer (IGV). A blacklist of over- and under sampled regions was generated by scoring read coverage of input and Col-0 ChIP samples across 200 bp windows using BEDtools [37]. Windows that were statistical outliers were determined using R and subsequently removed from the analysis. EPIC2 was used to determine enriched regions in two replicates against the pool of two Col-0 control IPs using pooled input samples as correction [38]. Replicates were compared using the Irreproducible Discovery Rate (IDR) framework [39]. Peak passing the threshold of 0.01 > IDR were merged using bedtools. Previously generated 35Sp-TRB1-YFP reads were included in the IDR analysis for better comparison [9].

## Analysis of the binding behavior of various epigenetic regulatory complexes

Binding peaks of UBP5, EPRC1, PWWP1, HAM1, and EPL1B were sourced from [11], CLF and SWN from [21], and JMJ14 and a second set of TRB1-3 binding data were obtained from [23]. To visualize the overlapping binding sites, the TAIR10 genome of *A. thaliana* was tiled into 397160 bins of 300 bp length and all peaks of these eight datasets and TRB1, TRB2 and TRB3 were assigned to overlapping bins. As the datasets were derived using different ChIP-Seq pipelines, the "Score" column of each dataset was normalised into deciles. Preliminary analysis of the overlap of the ChIP-Seq sets was performed through pearson correlation using Hmisc [40].

To visualize the overlapping binding sites, the bins were assigned to distinct categories: Bins bound by at least two of the PEAT components, both NuA4 components, both PRC2 components, or JMJ14 were assigned to "PEAT", "NuA4", "PRC2", or "JMJ14" respectively. In addition, each of the 12 possible combinations of multiple complex assignments were added along with a category for unassigned bins, bringing the total to 17 categories. Each bin was assigned to one of these categories. Statistical analysis was performed using the MSET function of the SuperExactTest package [25] for pairwise comparison of the overlap of the generated categories with TRB1,2,3-bound bins. Heatmaps were generated using the ComplexHeatmap [41] package.

## Cis-motif enrichment analysis

DNA motifs enriched in peak-assigned bins were identified through the XSTREME pipeline of the MEME-suit [42] using standard settings except for *--meme-mod "anr"*, providing the DNA motifs identified by [43]. Motifs were declared as telo-box-like, if the identified motif was closely related to the telo-box, but did not fully capture the canonical *Arabidopsis* telomere repeat sequence of TTTAGGG. For each peak category, motifs were ranked based on their e-Value.

## Gene Ontology enrichment analysis

The peaks of all ChIP-Seq sets used in this study were annotated to genes using "annotatePeak" of the "ChIPseeker" package [44,45]. Since epigenetic regulatory complexes are not solely found at or near the TSS, the parameters "tss-Region" and "overlap" were set to "c(-2000, 2000)" and "all" respectively in order to correctly assign binding sites further away from the TSS. Additionally, the wide nature of peaks derived from epigenetic regulatory complexes leads to a high proportion of multi-gene spanning peaks. To compensate for this, addFlankGeneInfo was set to true and all flanking genes with "flank_gene_distances" of 0 were also annotated to the same peak. The resulting genes were assigned to epigenetic regulatory complexes in the same manner as the genomic bins. The different sets were subsequently used to calculate GO-Term enrichment using the "clusterProfiler" package [46].

PLOS Genetics

## Sample preparation for LC-MS/MS

Leaf tissue (7g) harvested from 5-week-old transgenic plants (*CaMV 35Sp-TRB1-GFP*, *CaMV 35Sp-TRB3-GFP*, *CaMV 35Sp-EDS1-GFP*) was cut with scissors into 0.5 - 1.0 cm pieces and disrupted on ice in 15 ml Precellys tubes containing 5 ml extraction buffer (2 M hexylene glycol, 0.5 M PIPES-KOH pH7.0, 10 mM MgCl2, 5 mM beta-mercaptoethanol) and 13–15 sterilized metal beads using a Precellys 24 homogenizer (Bertin instruments) for three rounds set to 10 s at 7500 rpm. Samples were filtered through a single and then a double Miracloth (Merk) layer, adjusted to a volume of 45 ml. 10% Triton X-100 was added stepwise to a final concentration of 0.8%. While samples were incubated on ice, the Percoll (Sigma-Aldrich) gradient was assembled by carefully underlying 6 ml of 30% Percoll solution with 6 ml of 80% Percoll in a centrifuge tube (Beckman Coulter #355631). In parallel, three 15 ml aliquots per sample were layered onto gradients and centrifuged (2,000 g, 4 °C, 30 min). The nuclei-enriched fractions (5ml) were collected from the interphase between the Percoll layers using a 5-ml pipette and the combined aliquots diluted in 23 ml gradient buffer (0.09 M hexylene glycol, 0.09 mM PIPES-KOH pH7, 1.83 mM MgCl2, 0.92 mM β-mercaptoethanol, 0.18% Triton X-100). To gently pellet the nuclei, the samples cushioned on 6 ml 30% Percoll solution were centrifuged at 2,000 g and 4°C for 10 min. The isolated nuclei were resuspended in 1 ml sample buffer (20 mM TrisHCl pH7.4, 2 mM MgCl2, 150 mM NaCl, 5% glycerol, 5 mM DTT, complete protease inhibitor (Roche)) and transferred into fresh 1.5 ml Eppendorf tubes and once washed in sample buffer (cenrifugation 1,000 g at 4°C for 15 min) and resupended in a final volume of 600µl sample buffer. Samples were treated with 1 µl DNase I (10 u/µl) and 2 µl of RNase A (10 mg/ml) for 15 min at 37°C and subsequently sonicated in a Bioruptor (Diagenode) water bath connected to a Minichiller cooling system (Huber) (6x 15 s "on"/15 s "off" at high intensity). After removal of debris (centrifugation at 16,000 g and 4°C for 15 min), supernatants were transferred into clean 2 ml Protein LoBind tubes (Eppendorf). The protein concentration was determined by Bradford assay (Bradford, 1976) and equal amounts (i.e., 1 mg) were used for subsequent affinity purification. Immunoprecipitation was carried out with 25 µl GFP-trap Agarose beads (gta-20; Chromotek) in 2 ml sample buffer with Triton X-100 (0.1%) and EDTA (2 mM) after incubation at 4°C for 2.5 h at constant rotation (12 rpm). The protein-bound GFP-trap beads were washed four times with 300 µL of wash buffer (20 mM Tris-HCl pH7.4, 150 mM NaCl, 2 mM EDTA).

## Sample preparation and LC-MS/MS data acquisition

Proteins from GFP-trap enrichment were submitted to an on-bead digestion. In brief, dry beads were re-dissolved in 25 µL digestion buffer 1 (50 mM Tris, pH 7.5, 2M urea, 1mM DTT, 5 ng/µL trypsin) and incubated for 30 min at 30 °C in a Thermomixer with 400 rpm. Next, beads were pelleted, and the supernatant was transferred to a fresh tube. Digestion buffer 2 (50 mM Tris, pH 7.5, 2M urea, 5 mM CAA) was added to the beads; after mixing, the beads were pelleted, the supernatant was collected and combined with the previous one. The combined supernatants were then incubated o/n at 32 °C in a Thermomixer with 400 rpm; samples were protected from light during incubation. The digestion was stopped by adding 1 µL TFA and desalted with C18 Empore disk membranes according to the StageTip protocol [47]. Dried peptides were re-dissolved in 2% ACN, 0.1% TFA (10 µL) for analysis and measured without dilution. Samples were analyzed using an EASY-nLC 1200 (Thermo Fisher) coupled to a Q Exactive Plus mass spectrometer (Thermo Fisher). Peptides were separated on 16 cm frit-less silica emitters (New Objective, 75 µm inner diameter), packed in-house with reversed-phase ReproSil-Pur C18 AQ 1.9 µm resin (Dr. Maisch). Peptides were loaded on the column and eluted for 115 min using a segmented linear gradient of 5% to 95% solvent B (0 min: 5%B; 0–5 min -> 5%B; 5–65 min -> 20%B; 65–90 min -> 35%B; 90–100 min -> 55%; 100–105 min -> 95%, 105–115 min -> 95%) (solvent A 0% ACN, 0.1% FA; solvent B 80% ACN, 0.1%FA) at a flow rate of 300 nL/min. Mass spectra were acquired in data-dependent acquisition mode with a TOP15 method. MS spectra were acquired in the Orbitrap analyzer with a mass range of 300–1750 m/z at a resolution of 70,000 FWHM and a target value of $3 \times 10^6$ ions. Precursors were selected with an isolation window of 1.3 m/z. HCD

fragmentation was performed at a normalized collision energy of 25. MS/MS spectra were acquired with a target value of $10^5$ ions at a resolution of 17,500 FWHM, a maximum injection time (max.) of 55 ms and a fixed first mass of m/z 100. Peptides with a charge of +1, greater than 6, or with unassigned charge state were excluded from fragmentation for $MS^2$, dynamic exclusion for 30s prevented repeated selection of precursors.

### LC-MS/MS data data analysis

Raw data were processed using MaxQuant software (version 1.6.3.4, http://www.maxquant.org/) [48] with label-free quantification (LFQ) and iBAQ enabled [49]. MS/MS spectra were searched by the Andromeda search engine against a combined database containing the sequences from *A. thaliana* (TAIR10_pep_20101214; ftp://ftp.arabidopsis.org/home/tair/Proteins/TAIR10_protein_lists/) and sequences of 248 common contaminant proteins and decoy sequences. Trypsin specificity was required and a maximum of two missed cleavages allowed. Minimal peptide length was set to seven amino acids. Carbamidomethylation of cysteine residues was set as fixed, oxidation of methionine and protein N-terminal acetylation as variable modifications. Peptide-spectrum-matches and proteins were retained if they were below a false discovery rate of 1%.

Statistical analysis of the MaxLFQ values was carried out using Perseus (version 1.5.8.5, http://www.maxquant.org/). Quantified proteins were filtered for reverse hits and hits "identified by site" and MaxLFQ values were log2 transformed. Missing values were imputed from a normal distribution (1.8 downshift, separately for each column). After grouping samples by condition, only proteins with three valid values in at least one condition were retained for subsequent analysis. Statistically significant enrichment was performed by ANOVA followed by Honest True Difference (HSD) test for groups TRB1, TRB3, ESD1 with FDR < 0.05.

### Protein expression and purification

A single colony of *E. coli* SoluBL21 (amsbio), carrying either *pET-28b-TRB1 or pET-28b-TRB3* was used to inoculate 5ml preculture in LB-AMP (100 mg/ml ampicillin), and grown at 37°C, 200rpm overnight. The preculture was added to 1l LB-AMP-medium and grown at 37°C, 200 rpm until the $OD_{600}$ was around 0.6 – 0.8. After addition of 1mM IPTG the culture was transferred to 16°C,200rpm overnight. Bacterial cells were collected using a JLA 10.500 rotor (Beckman/centrifuge Avanti J-25) at 4000rpm at 4 °C for 10 min. Afterwards, bacterial cells were resuspended in 40ml ice-cold lysis buffer (50mM $NaPO_4$, 300mM NaCl, 10mM imidazole, 0.1M PMSF at pH 7.5) and disrupted using sonication (Ultrasonic-Desintegrator, Branson) (Programme: Strength: 6, Duty cycle: 40, 3x 2min). The cell debris was removed using a JA 25.50 rotor (Beckman/centrifuge Avanti J-25) at 13000 rpm, 4 °C for 30 min. For affinity purification, 500µl of Ni-NTA Agarose beads (Qiagen) were washed three times with 5 ml lysis buffer, collected at 800g,4°C for 1 min and added to the cell lysate. After incubation at 4°C, 12rpm for 2h then beads were collected, transferred to a fresh 5ml Eppendorf tube and washed five times with 5ml washing buffer (50mM $NaPO_4$, 300mM NaCl, 20mM imidazole, pH 7.5) at 4°C, 12rpm for 5min. To elute proteins, the beads were incubated with 1.5ml elution buffer (50mM $NaPO_4$, 300mM NaCl, 250mM imidazole, pH 7.5) at 4°C, 12rpm for 2h. After collecting the beads at 4 °C, 800 g for 1 min, the supernatant was collected and dialysed in 500ml dialysis buffer (50mM $NaPO_4$, 300mM NaCl, pH 7.5) using Slide-A-Lyzer Dialysis Cassettes (10K MWCO, 3 mL, ThermoFisher Scientific) to remove the imidazole. Dialysed proteins were collected in Protein LoBind Tubes (Eppendorf) and kept at 4°C. Protein quantity was determined by Bio-Rad Protein Assay according to manufacturer's instructions. To check protein integrity, 1µg of protein was mixed 2x SDS-Loading buffer (126 mM Tris-HCl (pH 6.8), 20% glycerol, 4% SDS, 0.02% bromophenol blue), incubated at 95°C for 5min, and run in 1x TGS buffer (Bio Rad) on a 1.5mm, 12% SDS-PAGE at 100 V for approximately 1.5h. The SDS-PAGE was stained with Coomassie brilliant blue staining solution (1g Coomassie Brilliant Blue (Bio-Rad), 500ml MeOH, 100ml glacial acetic acid, 400ml $H_2O$) and de-stained with $H_2O$ overnight.

## Microscale thermophoresis (MST)

Forward and reverse 5'-Cy3-labelled oligonucleotides of 28 bp were ordered from SIGMA-ALDRICH. Sequences originating from the *SEP3* promoter region were Cy3-*proSEP3*-telo-box-Cy3: TTTAAATGT<u>TAGGGTTT</u>TTTGTAGGATT and Cy3-*proSEP3*-NonInter-Cy3: AAAAATATTTATATCACATCATTGTTAT). Two versions of the (C)RACCTA motif were Cy3-(C) AACCTAA-Cy3: CATCATGG<u>CAACCTAA</u>GGCTGGTACT AG and Cy3-(C)GACCTAA-Cy3: CATCATGG<u>CGACCTAA</u>GGCTG GTACTAG. A four-telo-box-repeat (R4) oligomer Cy3-R4-telo-box-Cy3: GGTTTAGGGTTTAGGGTTTAGGGTTTAG was published in [27]. Annealing was carried out in a heating block in dialysis buffer at a concentration of 10μM sense and anti-sense oligonucleotides by first incubating at 95°C for 15 min and subsequent slow cooling by switching off the heating block.

For all MST experiments, the Monolith NT.115 instrument (NanoTemper Technologies) and 1x dialysis buffer with Tween 20 (0.05%) and BSA (1.25 mg/ml) were used. Oligomer fluorescence intensity, absorption, and bleaching was tested with the instrument's green channel *via* the *Pretest* feature included in the machine's MO.Control software (NanoTemper Technologies). Samples were prepared according to the suggested protocol included in the software. Oligomer concentration was adjusted to obtain ≥200 fluorescent counts at a laser power ≤80%. These conditions were met at an oligomer concentration of 20nM and an IR-laser power of 60% for Cy3-*proSEP3*-telo-box-Cy3 and Cy3-*proSEP3*-NonInter-Cy3 and of 80% for Cy3-R4-telo-box-Cy3, Cy3-(C)AACCTAA-Cy3, and Cy3-(C)GACCTAA-Cy3. Afterward, general TRB-telo-box/telo-box-like element interaction and suitability of different capillaries was tested *via* the Binding Check feature and samples were prepared as suggested by the software. For this purpose, the highest possible protein concentration was mixed with 20nM of fluorescently labelled oligomers and incubated at RT and in the dark for 10 min. Afterward, the TRB-dsDNA mix was loaded onto Monolith NT.115 Premium Capillaries (NanoTemper Technologies) that prevented surface absorption, as TRB proteins tended to absorb to standard capillaries. Afterward, TRB-telo-box/telo-box-like element interaction was quantified by using the software's *Binding Affinity* feature. For this MST assay, a dilution series was prepared according to the software's instructions, using the beforehand determined laser powers and 20nM of oligomer mixed with 8.5μM to 260pM of TRB protein. The TRB-DNA mix was incubated at RT in the dark for 10 min before being loaded onto Premium capillaries. Each measurement was repeated at least three times. Binding curves were analysed and $K_D$ values were calculated with the MO.Affinity Analysis software (NanoTemper Technologies) according to the manufacturer's instructions.

Scripts for read processing, alignments, and downstream data analysis and visualizations are available in Dryad (https://doi.org/10.5061/dryad.gtht76j2r) [50].

## Supporting information

**S1 Fig. DEGs detected between double and single mutants.**
(PDF)

**S2 Fig. Expression of flowering time pathway genes in *trb* mutants.**
(PDF)

**S3 Fig. Bioinformatics pipeline for ChIP-seq analysis and ChIP-seq quality control.**
(PDF)

**S4 Fig. Characterization of TRB1-3 ChIP-seq data.**
(PDF)

**S5 Fig. TRB target and interaction partner correlation analysis using an independent dataset for TRB1, TRB2 and TRB3.**
(PDF)

 

**PLOS Genetics**

**S6 Fig. Gene Ontology enrichment analysis.**
(PDF)

**S1 File. Details of RNAseq analysis.** Tab sig_1_wt_inS: DEGs trb1-2 vs Col-0. Tab sig_2_wt_inS: DEGs trb2-2 vs Col-0. Tab sig_2_wt_inS: DEGs trb3-2 vs Col-0. Tab sig_12_wt_inS: DEGs trb1-2 trb2-3 vs Col-0. Tab sig_13_wt_inS: DEGs trb1-2 trb3-2 vs Col-0. Tab sig_23_wt_inS: DEGs trb2-3 trb3-2 vs Col-0. Tab sig_12_1_inS: DEGs trb1-2 trb2-2 vs trb1-2. Tab sig_12_2_inS: DEGs trb1-2 trb2-2 vs trb2-2. Tab sig_13_1_inS: DEGs trb1-2 trb3-2 vs trb1-2. Tab sig_13_3_inS: DEGs trb1-2 trb3-2 vs trb3-2. Tab sig_23_2_inS: DEGs trb2-2 trb3-2 vs trb2-2. Tab sig_23_3_inS: DEGs trb2-2 trb3-2 vs trb3-2. Tab gene_categories: DEG categories for all conditions. Tab PAMk5: row scaled vst transformed count matrix across all genotypes for all DEGs with PAMclusters indicated. Tab detected_AGIs: all genes with acceptable read counts (sum > 50 counts across all samples).
(XLSX)

**S2 File. Details of ChIP-Seq analysis.** Tab peaks_TRB1: Annotated ChIP-Seq Peaks of TRB1:YFP. Tab peaks_TRB2: Annotated ChIP-Seq Peaks of TRB2:YFP. Tab peaks_TRB3: Annotated ChIP-Seq Peaks of TRB3:YFP.
(XLSX)

**S3 File. Details of genomic binding analysis.** Tab final_database_bins: Table of 11 regulatory protein occupancy of 300 bp wide genomic bins, assigned regulatory complexes, and combination of TRB paralogs. Tab final_database_genes: Table of 11 regulatory protein occupancy of genes, associated RNAseq Cluster, assigned regulatory complexes, and combination of TRB paralogs.
(XLSX)

**S4 File. Details of IP-MS-MS analysis.** Tab raw: MaxQuant result. Tab ANOVA_HSD_full: log2 transformed, filtered and imputed LFQ values, result of ANOVA and HSD annotated to all protein. Tab: Significant: All proteins part of significant HSD groups.
(XLSX)

**S5 File. Details of GO-Term enrichment analysis.** Tab All_GO_Terms: Table containing all enriched (p<=0.05) GO-terms, their associated complexes, and co-bound TRB-paralogs.
(XLSX)

## Author contributions

**Conceptualization:** Maik Mendler, Kristin Krause, Franziska Turck.

**Data curation:** Maik Mendler, Prathamesh Sannak, Sara Stolze, Hirofumi Nakagami.

**Formal analysis:** Maik Mendler, Kristin Krause, Prathamesh Sannak, Franziska Turck.

**Investigation:** Maik Mendler, Kristin Krause, Simone Zündorf, Petra Tänzler, Sara Stolze.

**Methodology:** Sara Stolze, Hirofumi Nakagami.

**Project administration:** Hirofumi Nakagami, Franziska Turck.

**Supervision:** Hirofumi Nakagami, Franziska Turck.

**Validation:** Hirofumi Nakagami.

**Visualization:** Maik Mendler.

**Writing – original draft:** Maik Mendler, Kristin Krause.

**Writing – review & editing:** Maik Mendler, Franziska Turck.

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
