## [Decision Letter · Decision Letter 0]

22 Jan 2026

PGENETICS-D-25-01278

Epigenetic gene regulation is controlled by distinct regulatory complexes utilizing specialized paralogs of TELOMERE REPEAT BINDING FACTORS

PLOS Genetics

Dear Dr. Turck,

Thank you for submitting your manuscript to PLOS Genetics. After careful consideration, we feel that it has merit but does not fully meet PLOS Genetics's publication criteria as it currently stands. Therefore, we invite you to submit a revised version of the manuscript that addresses the points raised during the review process.

Please submit your revised manuscript within 3 months. If you will need more time than this to complete your revisions, please reply to this message or contact the journal office at plosgenetics@plos.org. Please include the following items when submitting your revised manuscript:

We look forward to receiving your revised manuscript.

Kind regards,

Bao-Cai Tan

Academic Editor

PLOS Genetics

Marnie Blewitt

Section Editor

PLOS Genetics

Aimée Dudley

Editor-in-Chief

PLOS Genetics

Anne Goriely

Editor-in-Chief

PLOS Genetics

**Additional Editor Comments:**

Thanks for submitting your manuscript to PLoS Genetics. It has been reviewed by three experts in this field. All reviewers consider that the data in this manuscript are of high quality, the conclusions are well supported, and the findings provide insights into the mechanism underlying TRBs' target gene selection. However, the reviewers also raised some concerns and questions, which I believe will improve the quality and strength of this manuscript. Please consider those, and we look forward to considering a revised manuscript with all the issues addressed.

**Journal Requirements:**

https://journals.plos.org/plosgenetics/s/submission-guidelines#loc-parts-of-a-submission

- ® on pages: 15, and 19

- TM on page: 21.

5) We notice that your supplementary Figures are included in the manuscript file. Please remove them and upload them with the file type 'Supporting Information'. Please ensure that each Supporting Information file has a legend listed in the manuscript after the references list.

Potential Copyright Issues:

i) Please confirm (a) that you are the photographer of 1A, or (b) provide written permission from the photographer to publish the photo(s) under our CC BY 4.0 license.

ii) Figure 6C. Please confirm whether you drew the images / clip-art within the figure panels by hand. If you did not draw the images, please provide (a) a link to the source of the images or icons and their license / terms of use; or (b) written permission from the copyright holder to publish the images or icons under our CC BY 4.0 license. Alternatively, you may replace the images with open source alternatives. See these open source resources you may use to replace images / clip-art:

7) In the online submission form, you indicated that your data will be submitted to a repository upon acceptance. We strongly recommend all authors deposit their data before acceptance, as the process can be lengthy and hold up publication timelines. Please note that, though access restrictions are acceptable now, your entire minimal dataset will need to be made freely accessible if your manuscript is accepted for publication. This policy applies to all data except where public deposition would breach compliance with the protocol approved by your research ethics board. If you are unable to adhere to our open data policy, please kindly revise your statement to explain your reasoning and we will seek the editor's input on an exemption.

8) Please amend your detailed Financial Disclosure statement. This is published with the article. It must therefore be completed in full sentences and contain the exact wording you wish to be published.

9) Kindly revise your competing statement in the online submission form to align with the journal's style guidelines: 'The authors declare that there are no competing interests.'

**Reviewers' comments:**

Reviewer's Responses to Questions

**Comments to the Authors:**

Reviewer #1: Three TRB proteins have been identified as shared components of several chromatin-associated complexes, including PEAT, PRC2, and NAC-JMJ14. These TRB proteins were previously thought to function redundantly within these complexes to regulate chromatin status and transcription. Interestingly, this study reveals non-redundant roles for the TRB proteins, providing important insights into how they exert specialized functions in chromatin regulation and transcriptional control. In this study, integrated analyses based on genetic, RNA-seq, and ChIP-seq data were performed. The main conclusions are well supported by the results presented in the manuscript. However, because the phenomena described are complex, the presentation of the results is somewhat difficult to follow. I suggest that the authors describe the results in a more concise and streamlined manner to improve clarity and accessibility for a broader readership. Additional comments are provided below:

Major points:

1. It is interesting to find that the prope triple mutants that segregated functional alleles of TRB2 showed a significant early-flowering phenotype. Because TRB proteins have been shown to form a protein complex with JMJ14 and NAC050/052, this observation is consistent with prior reports describing early-flowering phenotypes in nac050/052 and jmj14 mutants. This needs to be indicated in the manuscript.

2. In line 182, the text indicates that "Obvious positive correlation was also observed between PEAT, NuA4, TRB1 and TRB2 and between TRB3 and PRC2". However, TRB3 shows a relatively high positive correlation with TRB-JMJ14 components, but shows low positive correlation levels with all other components. The correlation levels between TRB3 and PEAT components (PWWP1 and EPCR1) are even higher than the correlation between TRB3 and PRC2 components (CLF and SWN). The correlation between TRB3 and PRC2 requires cautious interpretation and should not be overstated.

3. The pairwise Pearson correlation matrix shown in Figure 2A is important for understanding the non-redundant functions of different TRB proteins. One confusing observation is that the ChIP-seq signals of PRC2 components (CLF and SWN) show only a weak negative correlation with those of the NuA4 component EPL1B. This analysis therefore appears insufficient to clearly distinguish repressive chromatin regions marked by PRC2 from active chromatin regions marked by NuA4. This issue should be resolved in the revised manuscript.

4. The DNA-binding affinities of TRB1 and TRB3 to different DNA fragments were compared, showing that the overall binding affinity of TRB1 was lower than that of TRB3, even for the R4 probe (Figure 4C). Notably, the purified TRB1 protein appears to be of lower quality than the TRB3 protein, as shown in Figure 4B. Therefore, the quality of the MST data should be improved to more convincingly confirm the differences in DNA-binding affinities between TRB1 and TRB3.

Minor points:

1. In line 65 and 66, "HAM1 and HAM2 which links PEAT to the NuA4 complex". This statement may cause misunderstanding, because PEAT and NuA4 complexes are independent complexes, and HAM1 and HAM2 are shared subunits of these complexes.

2. Figure 4A contains words with too small font, and the resolution of the figure is low. Please increase the size and resolution.

3. In line 227, "Figure 4" should be changed to "Figure 4A". The results described in line 235- 248 should contain referenced figures.

4. In line 271, "Figure 4G and G" should be changed to "Figure 4F and G".

5. The resolution of images shown in Figure 4A and Figure 6C should be improved, and the front size should be increased.

6. In line 227, replace the citation "Figure 4" with "Figure 4A".

7. In line 271, correct the citation "Figure 4G and G" to "Figure 4F and G".

Reviewer #2: In this manuscript, Maik Mendler and colleagues have attempted to address the redundancy between three telomere repeat binding transcription factors in target loci selectin and protein interactions. Redundancy among family members is a long-standing obstacle in understanding the function of each member. By using traditional forward and reverse genetic methods, the diversity among family genes is usually not be fully addressed due to the functional redundancy. In this work, the authors incorporated multiple genome-wide data and various bioinformatic methods to dissect the function of three TRB TFs, the DEG, the target selection, the associated regulatory protein complexes, and most importantly, the distinct biological function. The paper is well written, with rational and line of reasoning. The work is genuinely novel, orginal, and would provide insightful methodology of the field.

Specific Comments:

In the first part of the result section, the RNA-seq was conducted in single, double and triple mutants of trbs, and performed PAM clustering to assemble the genes into 4 groups. It would be better if DEGs were separated into a up and down genes first to give a direct view to the role of TRB in gene regulation.

In target gene part, the TRB1 targets were derived from published dataset, which showed a greater number of targets compared to TRB2 and TRB3. How could the authors prove that this difference is due to the character of the TFs but not the different conditions in generating the data.

Supplemental Fig 3B showed the comparison of number of peaks, but not the target, which is not effective for the comparison between their data and published data. In addition, it is better to give the general metaplot of the thee TRBs first, to give a comprehensive view of the genome-wide binding.

Line 177, the bin was 500 bp in the analysis. However, since the authors tried to associate TRBs with histone modifiers, the 500 bp bin is roughly too large, which spanned more than three nucleosomes. The authors could consider using a narrow bin within two nucleosomes for the correlation analysis.

P7 line 206, the IP-MS conducted with TRB1 and TRB3 and TRB2 was missing. Moreover, the genes were overexpressed, but TRB1 and TRB3 were quantitively compared. The over amount of TRB proteins may obscure the quantitative comparative analysis of their interactors.

For the MST analysis, TRB2 was missed.

Lastly, the biological function and targets of each TRB1 with their distinct regulatory complex were assigned. It was revealed that a specific TRB sites could be associated with several complexes. What experimental strategies could the authors adopt to validate the bioinformatic analysis? This is quite interesting, since the conclusion were based on multiple bioinformatic methods and co-operation of a single locus are usually observed for plants to tackle with various environmental stimuli and internal signal. Experimental validation will not only enhance the robustness of this work, but also shed light to the field.

Reviewer #3: Mendler et al. provide comprehensive RNA-seq and ChIP-seq data to dissect the target gene selection among TRB1, TRB2, and TRB3. Comparing the target sites of TRB proteins with those of a set of epigenetic regulators suggests that TRB proteins may target genes by associating with distinct epigenetic complexes. For example, the binding sites of TRB1 and TRB3 are strongly associated with the PEAT complex and PRC2, respectively. The interacting proteins co-purified with TRB1 and TRB3 further support this notion, as the PEAT complex subunits EPCR1 and EPCR2 are preferentially enriched by TRB1, while EMF1 is exclusively enriched by TRB3. Thus, this manuscript provides a mechanism for TRBs' target gene selection through preferential association with distinct epigenetic regulators.

Here, I list my concerns about this manuscript below:

1. The authors compared the transcriptome data among TRB single mutants and double mutants. It would be more appropriate to compare single mutants with their respective double mutants, such as comparing trb1, trb2, and trb1 trb2. Given that only about one-third of the differentially expressed genes (DEGs) identified in single and double mutants are shared with trb123, the authors should discuss this point.

2. In line 115, these mutants should be specified as trb123.

3. The binding sites of TRB proteins far exceed the DEGs; therefore, the relationship between TRB-regulated genes and TRB binding sites should be discussed.

4. The authors selected CLF and SWN binding sites as representatives of PRC2 targets [1]. However, these binding sites are significantly fewer than the total number of PRC2 target genes. Only a small fraction of genes exhibit reduced H3K27me3 in their mutants, making it inappropriate to select these datasets to represent PRC2 targets. Utilizing the FIE binding sites may provide a better alternative [2].

5. In lines 265-266, the rationale for choosing the SEP3 promoter should be clearly presented.

1. Shu J, Chen C, Thapa RK, Bian S, Nguyen V, Yu K, et al. Genome-wide occupancy of histone H3K27 methyltransferases CURLY LEAF and SWINGER in Arabidopsis seedlings. Plant Direct. 2019;3(1):e00100. doi: 10.1002/pld3.100. PubMed PMID: 31245749; PubMed Central PMCID: PMCPMC6508855.

2. Zhou Y, Wang Y, Krause K, Yang T, Dongus JA, Zhang Y, et al. Telobox motifs recruit CLF/SWN-PRC2 for H3K27me3 deposition via TRB factors in Arabidopsis. Nature genetics. 2018;50(5):638-44. doi: 10.1038/s41588-018-0109-9. PubMed PMID: 29700471.

**Have all data underlying the figures and results presented in the manuscript been provided?**

Large-scale datasets should be made available via a public repository as described in the *PLOS Genetics*
data availability policy, and numerical data that underlies graphs or summary statistics should be provided in spreadsheet form as supporting information., and numerical data that underlies graphs or summary statistics should be provided in spreadsheet form as supporting information., and numerical data that underlies graphs or summary statistics should be provided in spreadsheet form as supporting information., and numerical data that underlies graphs or summary statistics should be provided in spreadsheet form as supporting information.

Reviewer #1: None

Reviewer #2: None

Reviewer #3: Yes

PLOS authors have the option to publish the peer review history of their article (what does this mean?). If published, this will include your full peer review and any attached files.). If published, this will include your full peer review and any attached files.). If published, this will include your full peer review and any attached files.). If published, this will include your full peer review and any attached files.

...

Reviewer #1: No

Reviewer #2: No

Reviewer #3: No

**Figure resubmission:**
---

## [Decision Letter · Decision Letter 1]

31 Mar 2026

Dear Dr. Turck,

We are pleased to inform you that your manuscript entitled "Epigenetic gene regulation is controlled by distinct regulatory complexes utilizing specialized paralogs of TELOMERE REPEAT BINDING FACTORS" has been editorially accepted for publication in PLOS Genetics. Congratulations!

Yours sincerely,

Bao-Cai Tan

Academic Editor

PLOS Genetics

Marnie Blewitt

Section Editor

PLOS Genetics

Aimée Dudley

Editor-in-Chief

PLOS Genetics

Anne Goriely

Editor-in-Chief

PLOS Genetics

BlueSky: @plos.bsky.social

Comments from the reviewers (if applicable):

All the concerns have been addressed, and the conclusion is adequately supported by the data. I am pleased to inform you that this manuscript has been accepted for publication in PLOS Genetics.

Reviewer's Responses to Questions

**Comments to the Authors:**

Reviewer #1: All my comments have been well addressed in the revised manuscript. This manuscript reports the non-redundant roles of the three TRB proteins, which is important for understanding their complex biological functions in plants. I am satisfied with the manuscript, and have no further comments.

Reviewer #2: I consider the revised manuscript to be substantially improved, and I am pleased that the authors found merit in my suggestions. Overall, I find the work of sufficient quality for publication, as the main conclusions are adequately supported.

Reviewer #3: All my concerns have been satisfactorily addressed.

**Have all data underlying the figures and results presented in the manuscript been provided?**

Large-scale datasets should be made available via a public repository as described in the *PLOS Genetics*
data availability policy, and numerical data that underlies graphs or summary statistics should be provided in spreadsheet form as supporting information., and numerical data that underlies graphs or summary statistics should be provided in spreadsheet form as supporting information., and numerical data that underlies graphs or summary statistics should be provided in spreadsheet form as supporting information., and numerical data that underlies graphs or summary statistics should be provided in spreadsheet form as supporting information.

Reviewer #1: None

Reviewer #2: None

Reviewer #3: Yes

PLOS authors have the option to publish the peer review history of their article (what does this mean?). If published, this will include your full peer review and any attached files.). If published, this will include your full peer review and any attached files.). If published, this will include your full peer review and any attached files.). If published, this will include your full peer review and any attached files.

...

Reviewer #1: No

Reviewer #2: **Yes:** Zicong LiZicong LiZicong LiZicong Li

Reviewer #3: No

**Data Deposition**

If you have submitted a Research Article or Front Matter that has associated data that are not suitable for deposition in a subject-specific public repository (such as GenBank or ArrayExpress), one way to make that data available is to deposit it in the Dryad Digital Repository. As you may recall, we ask all authors to agree to make data available; this is one way to achieve that. A full list of recommended repositories can be found on our . As you may recall, we ask all authors to agree to make data available; this is one way to achieve that. A full list of recommended repositories can be found on our . As you may recall, we ask all authors to agree to make data available; this is one way to achieve that. A full list of recommended repositories can be found on our . As you may recall, we ask all authors to agree to make data available; this is one way to achieve that. A full list of recommended repositories can be found on our website....

http://datadryad.org/submit?journalID=pgenetics&manu=PGENETICS-D-25-01278R1

Additionally, please be aware that our data availability policy requires that all numerical data underlying display items are included with the submission, and you will need to provide this before we can formally accept your manuscript, if not already present. requires that all numerical data underlying display items are included with the submission, and you will need to provide this before we can formally accept your manuscript, if not already present. requires that all numerical data underlying display items are included with the submission, and you will need to provide this before we can formally accept your manuscript, if not already present. requires that all numerical data underlying display items are included with the submission, and you will need to provide this before we can formally accept your manuscript, if not already present.

**Press Queries**

If you or your institution will be preparing press materials for this manuscript, or if you need to know your paper's publication date for media purposes, please inform the journal staff as soon as possible so that your submission can be scheduled accordingly. Your manuscript will remain under a strict press embargo until the publication date and time. This means an early version of your manuscript will not be published ahead of your final version. PLOS Genetics may also choose to issue a press release for your article. If there's anything the journal should know or you'd like more information, please get in touch via plosgenetics@plos.org....

---

## [Editor Report · Acceptance letter]

PGENETICS-D-25-01278R1

Epigenetic gene regulation is controlled by distinct regulatory complexes utilizing specialized paralogs of TELOMERE REPEAT BINDING FACTORS

Dear Dr Turck,

We are pleased to inform you that your manuscript entitled "Epigenetic gene regulation is controlled by distinct regulatory complexes utilizing specialized paralogs of TELOMERE REPEAT BINDING FACTORS" has been formally accepted for publication in PLOS Genetics! Your manuscript is now with our production department and you will be notified of the publication date in due course.

With kind regards,

Judit Kozma

PLOS Genetics

On behalf of:
